# SEVERING THE LINK: A UNIFIED ADVERSARIAL ATTACK ON IMAGE AND VIDEO MLLMS VIA GENERATIVE DISRUPTION

> **Note to Reviewers:** Revisions made during the rebuttal period, including new experimental results, analysis, and appendices, are highlighted in blue for your convenience.

## ABSTRACT

While Multimodal Large Language Models (MLLMs) demonstrate remarkable cross-modal reasoning, their core vision-language grounding mechanisms present critical vulnerabilities, particularly in complex video scenarios. We introduce **CAVALRY**, a unified framework for generating powerful adversarial attacks against both image and video MLLMs. Our approach introduces two key innovations: **(i)** a paradigm shift from conventional classification-boundary attacks to directly disrupting the generative process, realized through a novel loss that maximizes the likelihood divergence of the ground-truth response and severs the visual-linguistic link; and **(ii)** an efficient, progressive generator trained to produce spatiotemporally coherent perturbations for both dynamic videos and static images. Comprehensive evaluations on seven state-of-the-art MLLMs, including GPT-4.1, Gemini 2.0, and QwenVL-2.5, validate CAVALRY's superior performance. Our method outperforms the strongest baselines by an average of 22.8% on holistic video understanding benchmarks and extends this advantage to static images, proving 34.4% more effective than prior work. These results establish CAVALRY as a foundational framework for probing the adversarial robustness of the entire spectrum of modern MLLMs.

## 1 INTRODUCTION

The emergence of foundation models integrating vision and language capabilities has transformed artificial intelligence research. These Multimodal Large Language Models (MLLMs) have demonstrated remarkable cross-modal understanding and reasoning abilities across both static images and dynamic video streams (Achiam et al., 2023; Team et al., 2024; Chen et al., 2024b; Zhang et al., 2024; Li et al., 2024b; Abdin et al., 2024; Lin et al., 2024; Bai et al., 2025; Li et al., 2024a; Zhang et al., 2025a). As these models are increasingly deployed in safety-critical domains such as autonomous driving and security systems, their vulnerabilities pose significant real-world risks. For example, malicious actors could potentially circumvent content moderation systems by adding adversarial perturbations to harmful multimedia content such as violent, explicit, or inflammatory material. However, their vulnerabilities to adversarial attacks remain insufficiently explored. This gap stems from three key challenges: the complex vision-language grounding mechanisms, the difficulty of generating coherent perturbations across spatial and temporal dimensions, and the computational burden of processing video sequences.

Existing adversarial attack methods are inadequate for jointly handling image and video MLLMs. Video-centric attacks targeting unimodal classifiers cannot effectively disrupt the sophisticated vision-language reasoning mechanisms (Wei et al., 2020; 2022b; Chen et al., 2024a; Wei et al., 2022a;c; 2023; Chen et al., 2023; Gao et al., 2024), while attacks on static vision-language models (e.g., CLIP (Radford et al., 2021)) fail to capture crucial temporal dependencies (Zhang et al., 2022; Zhao et al., 2023; Lu et al., 2023; Yin et al., 2023; Zhang et al., 2025b; Huang et al., 2025). This

highlights the need for a unified framework that leverages the inference efficiency of generative paradigms (Poursaeed et al., 2018; Xiao et al., 2018; Zhang et al., 2025b) to produce effective adversarial examples for both images and videos.

To address these challenges, we propose **CAVALRY** (**C**ross-mod**A**l **L**anguage-**V**ision **A**dve**R**sarial **Y**ielding), a unified framework for generating transferable adversarial examples against MLLMs. **(i)** We introduce *generative likelihood divergence maximization*, moving beyond traditional feature-space manipulations. Our approach employs a semantic-visual loss function that operates through token-level adversarial manipulation in autoregressive generation, disrupting the connection between visual perception and language generation. **(ii)** We develop a *progressive two-stage cross-modal generator* for video attacks. The framework combines large-scale multimodal pre-training with specialized fine-tuning for spatiotemporal coherence. By capturing temporal correlations through structured batch processing rather than explicit regularization, our approach maintains flexibility for both video and image inputs without requiring specialized architectures.

Extensive experiments on both videos and images across seven state-of-the-art MLLMs, including commercial systems (GPT-4.1 (Achiam et al., 2023), Gemini 2.0 (Team et al., 2024)) and leading open-source models (QwenVL-2.5 (Bai et al., 2025), InternVL-2.5 (Chen et al., 2024b), Llava-Video (Zhang et al., 2024), Aria (Li et al., 2024b), MiniCPM-o-2.6 (Yao et al., 2024)). Our approach achieves state-of-the-art performance, with an average relative improvement of **22.8%** on the holistic video benchmark and **34.4%** on the image benchmark. The public release of our framework and model weights will facilitate broader research into MLLM robustness. Our primary contributions are threefold:

- We establish a novel paradigm for cross-modal adversarial attacks based on generative likelihood divergence maximization, targeting evidence-grounded language generation rather than classification boundaries.

- We design a progressive generator framework that can produce spatially and spatiotemporally coherent perturbations for both images and videos.

- We demonstrate the state-of-the-art effectiveness of our framework through extensive experiments on seven leading MLLMs across comprehensive video and image understanding benchmarks.

## 2 RELATED WORK

**Multimodal Large Language Models** The field of MLLMs evolved from foundational vision-language models like CLIP (Radford et al., 2021), which established crucial cross-modal alignment paradigms. Academic research has driven significant innovations through specialized architectures: InternVL (Chen et al., 2024b) and Qwen-VL series (Bai et al., 2025) pioneered efficient frame processing strategies, while LLaVA-Video (Zhang et al., 2024) advanced temporal reasoning through multi-stage training for video MLLMs. Open-source efforts like Phi-3.5 (Abdin et al., 2024), VILA (Lin et al., 2024), MiniCPM-o (Yao et al., 2024) and Aria (Li et al., 2024b) have demonstrated the effectiveness of extensive video-instruction tuning and novel temporal aggregation methods. Commercial models such as GPT-4 (Achiam et al., 2023) and Gemini (Team et al., 2024) represent the current state-of-the-art with their end-to-end architectures integrating perception and reasoning within unified generative frameworks.

**Multi-modal Adversarial Attacks** Research on adversarial attacks has made significant progress along two complementary directions, each providing valuable insights for our work. The first direction has explored video (pure-vision) model robustness through temporal modeling attacks that extend image-based adversarial techniques to video sequences on action recognition and classification tasks (Wei et al., 2020; 2022b; Chen et al., 2024a; Wei et al., 2022a;c; 2023; Chen et al., 2023; Gao et al., 2024; Luo et al., 2024; Cui et al., 2024). The second direction has explored adversarial vulnerabilities in VLMs, with recent work developing effective methods for disrupting image-text alignment in multimodal systems. These methods revealed important insights about the vulnerability of cross-modal integration mechanisms (Zhang et al., 2022; Zhao et al., 2023; Lu et al., 2023; Yin et al., 2023; Zhang et al., 2025b; Huang et al., 2025). However, with the rapid advancement of MLLMs, these methods face significant challenges in effectively handling video data. Among these methods,

generator-based approaches offer promising directions by enabling efficient adversarial example generation without requiring iterative PGD-like optimization for each frame. As a complementary approach, Universal Adversarial Perturbations (UAPs) (Moosavi-Dezfooli et al., 2017), particularly those designed for VLMs (Zhou et al., 2023; Fang et al., 2024a; Huang et al., 2025), present another promising solution due to their minimal computational overhead and applicability to video data. Building upon these valuable contributions, our CAVALRY framework addresses the unique challenges at their intersection: simultaneously modeling cross-modal reasoning vulnerabilities and temporal dependencies while maintaining computational feasibility and black-box transferability for real-world content.

## 3   PROBLEM FORMULATION

Let $V = \{x_1, x_2, ..., x_N\}$ represent a video with $N$ frames, where each frame $x_i \in \mathbb{R}^{C \times H \times W}$ is an image with $C$ channels and dimensions $H \times W$. In practice, a MLLM typically samples a subset of frames $S = \{x_{t_1}, x_{t_2}, ..., x_{t_k}\}$ where $k \ll N$ due to computational constraints. Notably, the frame sampling strategy $\phi : V \to S$ varies significantly across different MLLM architectures, with models employing uniform sampling, scene-change detection, or learned importance metrics. A target MLLM, denoted as $\mathcal{T}$, processes these sampled frames along with a question $Q$ to generate a textual answer $A$:

$$A = \mathcal{T}(\phi(V), Q). \tag{1}$$

Unlike traditional classification models that output probability distributions over discrete labels, MLLMs generate natural language responses that interpret visual content through complex cross-modal reasoning processes. For image inputs, the formulation simplifies by replacing $\phi(V)$ with a single image $x$ in Equation (1).

Given a video $V$, a question $Q$, and a ground-truth answer $A_{gt}$, we generate a perturbed video $V' = V + \delta = \{x'_1, x'_2, ..., x'_N\}$ on a white-box surrogate model $\mathcal{M}$, where $\delta = \{\delta_1, \delta_2, ..., \delta_N\}$ represents frame-wise perturbations. The attack aims to optimize the following objective:

$$\delta^* = \arg\max_{\delta} D(\mathcal{M}(\phi(V + \delta), Q), A_{gt}), \quad \text{s.t.} \quad \|\delta_i\|_\infty \le \epsilon, \tag{2}$$

where $D(\cdot, \cdot)$ is a semantic divergence function measuring the dissimilarity between the model's response to the perturbed video and the ground-truth answer. The perturbation is constrained to be visually imperceptible through the $\ell_\infty$-norm bound $\epsilon$. For image inputs (single frame), the formulation adapts by replacing $\phi(V + \delta)$ with $x + \delta$ in Equation (2), enabling our framework to address the broader spectrum of multimodal systems.

**Threat Model**   We focus on the challenging black-box transfer attack scenario, which represents the most realistic threat for deployed MLLMs. In this setting, the attacker generates perturbations using a white-box surrogate model $\mathcal{M}$ and evaluates their transferability to two types of black-box targets $\mathcal{T}$: (1) Open-source models deployed locally using official weights (e.g., Qwen-2.5, InternVL-2.5); (2) Commercial models accessible solely through their APIs (e.g., GPT-4.1, Gemini 2.0).

## 4   METHODOLOGY

**Framework Overview**   We introduce CAVALRY, a transferable adversarial attack framework designed to address the unique challenges posed by MLLMs. Figure 1 illustrates the pipeline of our approach. At its core, CAVALRY employs a UNet-style (Ronneberger et al., 2015) generator $G$ that transforms an input video $V \in \mathbb{R}^{N \times C \times H \times W}$ into restricted adversarial perturbations $\delta \in \mathbb{R}^{N \times C \times H \times W}$ (or an input image when $N = 1$), constrained such that $\|\delta\|_\infty \le \epsilon$ to ensure imperceptibility.

### 4.1   THEORETICAL FOUNDATION: FROM LIKELIHOOD DIVERGENCE TO SEMANTIC DISRUPTION

We first establish the theoretical connection between our optimization objective and the goal of creating semantically meaningful attacks. Unlike classification attacks that target decision boundaries, attacking MLLMs requires corrupting the entire generative process that translates visual evidence into text.

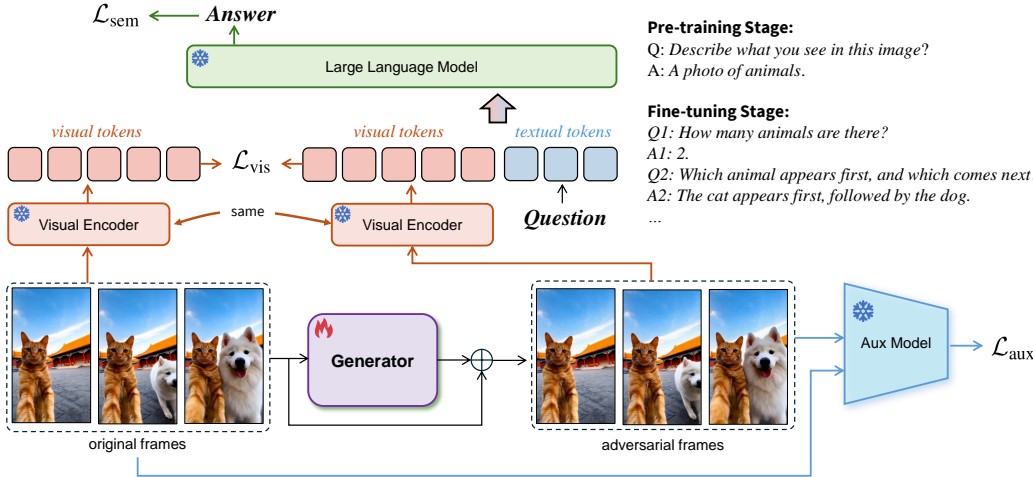

Figure 1: **Overview of the CAVALRY framework.** The generator $G$ produces perturbation patterns of identical dimensions to the input frames, and constitutes the only trainable component within the system, with all other network parameters remaining frozen. The colored arrows sequentially indicate the computation flow for three different objective functions. The training process is divided into two principal stages: (1) pre-training on a large-scale dataset with fixed `Question` templates to establish foundational adversarial patterns, and (2) fine-tuning with more diverse question-answer pairs and temporally correlated video data to enhance spatiotemporal coherence and attack transferability.

**Theorem 1** (Likelihood-KL Equivalence). *Maximizing the negative log-likelihood of the ground-truth answer (which we later formalize as $\mathcal{L}_{sem}$ in Section 4.2) is equivalent to maximizing the KL divergence between a Dirac delta distribution centered on the ground-truth answer and the model's predictive distribution under perturbed input.*

*Proof.* Let $P_{\mathcal{M}}(\cdot \mid V', Q)$ denote the model's distribution over answers given perturbed video $V'$, and $P_{gt}$ be the delta distribution at the ground-truth answer $A_{gt}$. The KL divergence is:

$$D_{\text{KL}}(P_{gt} \parallel P_{\mathcal{M}}(\cdot \mid V', Q)) = \sum_A P_{gt}(A) \log \frac{P_{gt}(A)}{P_{\mathcal{M}}(A \mid V', Q)}$$

$$= -\log P_{\mathcal{M}}(A_{gt} \mid V', Q). \tag{3}$$

For autoregressive generation, this expands to:

$$-\sum_{t=1}^{|A_{gt}|} \log P_{\mathcal{M}}(a_t \mid V', Q, a_{<t}), \tag{4}$$

which is precisely our semantic loss objective. $\square$

By Pinsker's inequality, the total variation distance (TV) between two distributions is upper-bounded by a function of their KL divergence ($D_{\text{TV}} \leq \sqrt{\frac{1}{2} D_{\text{KL}}}$). Thus, maximizing the KL divergence serves as a strong incentive for creating a substantial separation in output distributions. This theoretical insight motivates our dual-objective approach: *Semantic disruption* directly maximizes output distribution divergence, while *visual corruption* ($\mathcal{L}_{\text{vis}}$, detailed in Section 4.2) attacks the visual representations feeding the language model. The auxiliary loss $\mathcal{L}_{\text{aux}}$ further encourages perturbations to lie in vulnerability subspaces shared across architectures, enhancing transferability. By jointly optimizing these objectives, CAVALRY systematically dismantles the vision-language interface at its core.

## 4.2 DISRUPTING CROSS-MODAL INTEGRATION

Traditional adversarial attacks for image classification models typically target class boundaries or feature representations within a single modality. However, the power of MLLMs lies in their ability

to integrate visual perception with language understanding, a fundamentally different mechanism that requires a novel attack approach. Rather than manipulating classification logits, we implement generative likelihood divergence maximization to target the critical pathway through which visual information influences language generation, effectively severing the connection between what the model perceives and what it generates.

**Cross-Modal Disruption via Generative Likelihood Divergence**    Let $\mathcal{M}_{\text{LM}}$ represent the language generation component of a surrogate multimodal system and $\mathcal{M}_{\text{VE}}$ denote its visual encoding component. For given visual content $V$, question $Q$, and ground-truth answer $A_{gt}$, we formulate our semantic loss to directly disrupt language generation through generative likelihood divergence maximization, creating systematic deviations from expected cross-modal outputs:

$$\mathcal{L}_{\text{sem}}(\theta) = -\mathbb{E}_t \left[\log P_{\mathcal{M}_{\text{LM}}} \left(a_t | \mathcal{M}_{\text{VE}}(V + G_\theta(V)), Q, a_{<t}\right)\right], \tag{5}$$

where $a_t$ represents tokens in the ground-truth answer and $a_{<t}$ represents previously generated tokens. This approach implements token-level adversarial manipulation in autoregressive generation, directly compromising the model's ability to generate contextually appropriate responses based on visual evidence. Unlike traditional attacks that target intermediate representations, this function exploits temporal dependency vulnerabilities in sequential language generation conditioned on corrupted visual evidence.

**Visual Representation Manipulation**    To amplify the attack effectiveness, we simultaneously target the visual representation space through a complementary feature-level loss:

$$\mathcal{L}_{\text{vis}}(\theta) = \|\mathcal{M}_{\text{VE}}(V) - \mathcal{M}_{\text{VE}}(V + G_\theta(V))\|_2^2. \tag{6}$$

This component maximizes the mutual information gap between clean and perturbed cross-modal representations, creating significant shifts in visual encodings while maintaining imperceptibility in pixel space. While the semantic loss focuses on end-to-end generative process disruption, this complementary approach directly targets intermediate visual encodings that serve as critical inputs to cross-modal integration. By perturbing these representations, we corrupt the visual evidence before it reaches the reasoning components, creating cascading failures throughout the multimodal reasoning pipeline. To further ensure cross-architecture transferability, we incorporate an auxiliary feature loss using an adversarially trained auxiliary model $\mathcal{F}$:

$$\mathcal{L}_{\text{aux}}(\theta) = \|\mathcal{F}(V) - \mathcal{F}(V + G_\theta(V))\|_2^2 \tag{7}$$

This auxiliary component leverages an adversarially trained model with smoother feature spaces, enhancing transferability across different architectural choices.

The complete loss function, which we aim to maximize, combines these components to achieve comprehensive cross-modal disruption:

$$\mathcal{L}(\theta) = \lambda_1 \mathcal{L}_{\text{sem}}(\theta) + \lambda_2 \mathcal{L}_{\text{vis}}(\theta) + \lambda_3 \mathcal{L}_{\text{aux}}(\theta) \tag{8}$$

where $\lambda_1, \lambda_2, \lambda_3$ are hyperparameters balancing the relative importance of each loss component.

### 4.3    PROGRESSIVE TWO-STAGE GENERATOR TRAINING

To achieve spatiotemporal coherence in video attacks, we employ a two-stage training approach that progressively adapts the generator from broad cross-modal patterns to fine-grained spatiotemporal requirements. Once trained, the generator enables real-time adversarial attacks with minimal computational overhead during inference.

**Large-scale Pre-training**    We first pre-train our generator on the LAION-400M dataset (Schuhmann et al., 2021), which contains a vast collection of image-caption pairs spanning diverse visual concepts and domains. During this stage, we treat each image as individual visual input, construct a uniform question $Q$ = *"Describe what you see in this image"* for all samples, and use the image caption as the ground-truth answer $A_{gt}$. We optimize using the $\mathcal{L}(\theta)$ defined in Equation (8). This large-scale pre-training phase is critical as it enables the generator to learn diverse perturbation patterns that align with the knowledge distribution of potential target multimodal systems, establishing a strong foundation for cross-architecture transferability.

**Fine-tuning for Cross-Modal and Temporal Coherence** The foundation generator is further refined through progressive fine-tuning. First, we fine-tune on the LLaVA-Instruct-150K dataset (Liu et al., 2023), which provides diverse question-answering dialogues covering a wide range of visual reasoning tasks. We continue treating images as individual inputs while leveraging the dataset's diverse question types $Q$ and corresponding answers $A_{gt}$ via Equation (8). Finally, we validate and refine our approach on the most challenging multimodal scenario by fine-tuning on the Video-MME dataset (Fu et al., 2025) using the same objective function. The key distinction in this phase is our training approach: each batch contains frames sampled from the same video, sharing identical questions and ground-truth answers. This structured batching strategy implicitly encourages the generator to produce temporally coherent perturbations without requiring explicit temporal regularization terms. By seeing multiple frames from the same video context during training, the generator learns to produce consistent perturbations across a temporal sequence.

# 5 EXPERIMENTS

## 5.1 EXPERIMENTAL SETUP

**Evaluation Protocol** We evaluate CAVALRY on a diverse set of MLLMs, including five leading open-source models (Qwen-2.5, InternVL-2.5, Llava-Video, Aria, MiniCPM-o-2.6) and two commercial systems (GPT-4.1, Gemini 2.0). Our evaluation is primarily conducted on the MMBench-Video (Fang et al., 2024b) and MME (Fu et al., 2023) benchmarks using the VLMEvalKit[1]. Instead of simple binary accuracy, the native *Score Metric* of these modern benchmarks is a holistic quality score provided by a powerful LLM judge. A prime example is MMBench-Video (Fang et al., 2024b), which uses fine-grained Likert-style ratings for complex reasoning and then averages these scores across many questions. This evaluation paradigm inherently produces a *highly compressed and flattened score distribution* where absolute changes are minimal. Consequently, the resulting SRR values are on a different scale than traditional ASR: an SRR in the range of 5-15% signifies a substantial and effective degradation of model capabilities. We therefore define our primary metric, the **Score Reduction Rate (SRR)**, as:

$$\text{SRR} = 1 - (\text{Score}_{\text{attacked}}/\text{Score}_{\text{clean}}) \tag{9}$$

To provide a complementary assessment of fine-grained spatiotemporal reasoning, we additionally employ the **TempCompass** benchmark (Liu et al., 2024). For this specific task, we adopt the standard **Attack Success Rate (ASR)**, where a higher ASR indicates a more effective attack.

**Baselines and Implementation** We compare CAVALRY against a comprehensive suite of attacks: the single-modality GCMA (Chen et al., 2023), the video-adapted CWA (Chen et al., 2024a), the multimodal AnyAttack (Zhang et al., 2025b), and the universal X-Transfer (Huang et al., 2025). For our method, we use InternVL-2.5-1B as the surrogate model $\mathcal{M}$ with $\epsilon = 16/255$ and follow the proposed progressive training strategy. Key loss weights in Equation (8) were set to $\lambda_1 = 0.1$, $\lambda_2 = 20$, and $\lambda_3 = 10$. For complete reproducibility, all detailed configurations, including the specific LLM judges used for evaluation, baseline adaptations, and our full training hyperparameters, are provided in Appendix B.

## 5.2 VIDEO UNDERSTANDING RESULTS

We begin by evaluating CAVALRY against baseline attacks on the MMBench-Video benchmark to assess general video understanding capabilities across diverse perception and reasoning tasks. Table 1 presents the comparative results measured by Score Reduction Rate (SRR). CAVALRY consistently achieves the highest SRR across every tested model, demonstrating robust effectiveness in degrading overall system performance. In terms of general performance degradation, our method achieves an average relative improvement of 22.8% over the strongest competing baseline across all models. The performance gap is particularly pronounced for open-source models, where we observe substantial gains on InternVL2.5-8B (35.7% improvement), QwenVL-2.5-7B (33.3%), and MiniCPM-o-2.6 (44.4%). Furthermore, our approach maintains a measurable lead on highly robust commercial systems, improving attack effectiveness by 11.1% on Gemini-2.0-Flash and 5.9% on GPT-4.1-Mini.

---

[1] https://github.com/open-compass/VLMEvalKit

Table 1: Performance comparison on the MMBench-Video benchmark, measured by Score Reduction Rate (SRR %). Higher SRR indicates a more successful attack. Best results are in **bold**.

| Model | Attack | Overall SRR | Perception | | | | | Reasoning | | | | | |
|---|---|---|---|---|---|---|---|---|---|---|---|---|---|
| | | | CP | FP-S | FP-C | HL | Mean | LR | AR | RR | CSR | TR | Mean |
| *Open-Source MLLMs* | | | | | | | | | | | | | |
| InternVL 2.5-8B | GCMA | 1.2 | 1.7 | 1.7 | 4.7 | -16.1 | 1.8 | 1.0 | 1.9 | 0.0 | -0.7 | 2.9 | 1.4 |
| | CWA | 5.0 | 4.1 | 5.1 | 8.1 | 1.8 | 5.4 | -2.1 | 2.5 | 10.2 | 1.4 | 5.1 | 4.1 |
| | AnyAttack | 5.0 | 4.7 | 5.1 | 7.4 | 5.4 | 5.4 | -4.2 | 2.5 | 7.9 | 0.7 | 5.8 | 4.1 |
| | X-Transfer | 8.7 | 9.3 | 8.5 | 13.4 | 30.4 | 9.6 | -8.3 | 9.4 | 9.0 | 10.8 | 10.1 | 8.2 |
| | **CAVALRY** | **11.8** | 7.6 | 12.5 | 19.5 | 21.4 | **12.6** | -8.3 | 15.0 | 18.1 | 6.8 | 5.8 | **9.6** |
| MiniCPM o-2.6 | GCMA | 1.2 | 3.8 | 1.6 | 3.2 | 9.5 | 2.8 | -2.1 | 4.3 | -5.5 | -6.7 | 1.9 | -0.6 |
| | CWA | 5.2 | 8.8 | 5.3 | 7.6 | 13.1 | 6.7 | 4.1 | 7.9 | 1.8 | -3.9 | 1.9 | 2.5 |
| | AnyAttack | 5.2 | 8.8 | 5.9 | 8.9 | 11.9 | 7.3 | 2.7 | 7.9 | 0.0 | -3.9 | 2.6 | 2.5 |
| | X-Transfer | 5.2 | 7.1 | 4.8 | 7.0 | 22.6 | 6.7 | 3.4 | 9.1 | 5.5 | -9.6 | 3.2 | 3.8 |
| | **CAVALRY** | **7.6** | 7.1 | 8.0 | 12.1 | 13.1 | **8.9** | 2.7 | 11.6 | 1.8 | -1.7 | 0.0 | **3.1** |
| QwenVL 2.5-7B | GCMA | 1.4 | 2.4 | 2.8 | -3.6 | 0.0 | 2.1 | -3.8 | -1.9 | 7.7 | -7.4 | 1.7 | 0.7 |
| | CWA | 6.2 | 10.2 | 7.6 | 7.1 | -1.8 | 7.6 | 2.5 | 5.0 | 6.5 | -3.4 | 9.4 | 5.5 |
| | AnyAttack | 9.7 | 12.0 | 10.3 | 11.6 | 4.4 | 10.4 | 1.3 | 9.4 | 14.2 | -8.1 | 8.5 | 6.9 |
| | X-Transfer | 10.3 | 13.2 | 12.4 | 13.4 | 1.8 | 11.8 | -8.1 | 12.5 | 18.1 | -0.7 | 8.5 | 7.6 |
| | **CAVALRY** | **13.8** | 16.8 | 14.5 | 18.8 | -4.4 | **14.6** | -1.3 | 15.6 | 24.5 | -0.7 | 12.0 | **11.7** |
| Aria | GCMA | 0.6 | -2.3 | 0.6 | -5.3 | -8.0 | 0.6 | 0.0 | 1.1 | -2.4 | 1.7 | 4.7 | 1.2 |
| | CWA | 2.5 | 2.3 | 3.1 | -4.6 | -14.9 | 3.2 | -1.2 | 0.0 | 1.8 | 1.7 | 6.8 | 1.8 |
| | AnyAttack | 5.0 | 6.4 | 4.3 | 2.3 | 24.1 | 5.1 | 1.2 | 5.0 | 7.2 | 3.4 | 6.8 | 4.9 |
| | X-Transfer | 5.6 | 7.6 | 5.6 | 2.3 | 25.3 | 6.3 | 0.6 | 8.3 | 6.6 | -2.8 | 6.8 | 4.9 |
| | **CAVALRY** | **6.3** | 7.0 | 5.6 | 7.6 | 14.9 | **7.0** | 0.6 | 10.0 | 5.4 | 2.8 | 8.1 | **6.1** |
| Llava-Video 7B | GCMA | 1.3 | 3.5 | 0.0 | 1.4 | 30.4 | 1.2 | -0.8 | 4.2 | 0.0 | 7.6 | 1.5 | 2.7 |
| | CWA | 3.8 | 6.4 | 3.5 | 3.5 | 21.7 | 4.3 | -3.1 | 3.0 | 1.4 | 4.7 | 4.4 | 2.7 |
| | AnyAttack | 5.7 | 8.7 | 4.7 | 2.8 | 21.7 | 5.6 | -2.4 | 5.4 | 7.0 | 9.3 | 7.4 | 6.1 |
| | X-Transfer | 7.0 | 11.6 | 5.3 | 2.8 | 17.4 | 6.8 | -2.4 | 10.2 | 6.3 | 5.8 | 10.4 | 7.4 |
| | **CAVALRY** | **8.2** | 10.5 | 7.6 | 6.3 | 21.7 | **8.0** | 0.0 | 8.4 | 6.3 | 15.1 | 10.4 | **8.1** |
| *Commercial MLLMs* | | | | | | | | | | | | | |
| Gemini 2.0-Flash | GCMA | 6.9 | -1.6 | 12.1 | 2.6 | 40.0 | 9.0 | 15.4 | 3.3 | -8.6 | -6.4 | 4.5 | 1.4 |
| | CWA | 6.9 | 3.3 | 10.5 | -5.2 | 40.0 | 7.8 | -15.4 | 6.1 | 9.5 | -13.6 | 9.8 | 4.2 |
| | AnyAttack | 11.3 | 8.2 | 13.8 | 23.5 | -20.0 | 13.3 | -30.8 | 16.1 | 13.8 | -46.4 | 7.5 | 2.8 |
| | X-Transfer | 9.4 | 8.2 | 7.2 | 10.4 | 20.0 | 7.2 | 15.4 | 18.9 | 4.3 | -20.0 | -3.0 | 6.9 |
| | **CAVALRY** | **12.6** | 14.8 | 16.0 | 15.7 | 60.0 | **15.7** | -30.8 | 20.6 | 4.3 | -26.4 | 15.0 | **6.3** |
| GPT 4.1-Mini | GCMA | 4.3 | 6.3 | 8.0 | 6.2 | 0.0 | 6.2 | 0.0 | -1.1 | -2.5 | 3.6 | 9.8 | 0.0 |
| | CWA | 5.9 | 12.6 | 10.0 | -9.0 | 0.0 | 7.3 | 5.9 | 3.2 | 5.5 | 7.6 | 12.3 | 4.9 |
| | AnyAttack | 7.4 | 17.6 | 11.9 | -4.8 | -100.0 | 8.8 | -5.9 | 4.8 | 5.5 | 14.7 | 6.1 | 4.9 |
| | X-Transfer | 9.0 | 15.1 | 14.4 | -9.0 | 0.0 | 11.4 | -5.9 | 3.2 | 8.0 | 18.7 | 3.7 | 4.4 |
| | **CAVALRY** | **9.6** | 21.3 | 13.4 | -15.2 | 0.0 | **10.4** | 17.6 | 12.2 | 5.5 | 7.6 | 18.4 | **12.1** |

This consistent superiority across diverse architectures validates the strong transferability of our attack method on comprehensive video tasks.

**Evaluation on Temporal Robustness (TempCompass)** To rigorously assess CAVALRY's capability in disrupting fine-grained spatiotemporal reasoning, we extended our evaluation to the TempCompass benchmark (Liu et al., 2024). Unlike general video QA tasks, TempCompass isolates specific temporal attributes such as action, speed, direction, and order. Table 2 reports the Attack Success Rate (ASR), where a higher score indicates a more effective attack. The results unequivocally demonstrate the superiority of our framework: CAVALRY achieves the highest Overall ASR across all seven evaluated models, surpassing state-of-the-art baselines. Notably, on QwenVL-2.5-7B, our method reaches an ASR of 47.10%, establishing a significant lead over the strongest competitor (44.34%). Even on robust commercial systems like Gemini-2.0-Flash and GPT-4.1-Mini, our approach maintains a consistent advantage. This comprehensive success across diverse temporal dimensions confirms that our generative disruption effectively severs the spatiotemporal dependencies essential for video understanding.

## 5.3 IMAGE UNDERSTANDING RESULTS

As established in our problem formulation, CAVALRY naturally handles static images, enabling direct application to image understanding tasks without architectural modifications. We evaluate this capability on the MME image benchmark, with results measured in SRR presented in Table 3. The results demonstrate the broad effectiveness of our method, which achieves a state-of-the-art SRR on six out of the seven tested MLLMs. The only exception is Gemini-2.0-Flash, where the UAP-based X-Transfer method achieves a slightly higher SRR. To quantify our method's advantage where it excels, we find that its attack effectiveness, as measured by SRR, is on average **34.4% higher than that of the strongest competing baseline** across these six models. The performance

Table 2: Performance comparison on the TempCompass benchmark. We report the Attack Success Rate (ASR %), where a higher score indicates a more successful attack. Best Overall results are in **bold**. MC: Multi-Choice, Cap: Captioning, Cap-M: Caption Matching, Att: Attribute Change.

| Model | Attack | Overall ASR | Fine-grained Metrics (ASR %) | | | | | | | | |
|---|---|---|---|---|---|---|---|---|---|---|---|
| | | | MC | Yes/No | Cap | Cap-M | Action | Direct. | Speed | Order | Att |
| *Open-Source MLLMs* | | | | | | | | | | | |
| InternVL 2.5-8B | GCMA | 32.19 | 33.99 | 31.15 | 40.07 | 21.49 | 10.97 | 49.31 | 46.25 | 25.29 | 28.76 |
| | CWA | 32.27 | 33.99 | 31.15 | 40.37 | 21.49 | 11.09 | 49.43 | 46.51 | 25.00 | 28.91 |
| | AnyAttack | 35.66 | 38.04 | 34.12 | 43.86 | 24.75 | 15.28 | 50.88 | 48.27 | 29.70 | 34.09 |
| | X-Transfer | 37.75 | 40.82 | 35.39 | 45.41 | 28.14 | 18.55 | 52.76 | 50.68 | 31.79 | 34.66 |
| | **CAVALRY** | **41.43** | 43.61 | 38.65 | 52.30 | 29.21 | 23.04 | 54.46 | 48.66 | 38.80 | 42.61 |
| MiniCPM o-2.6 | GCMA | 39.96 | 41.01 | 31.63 | 47.55 | 42.32 | 15.40 | 56.22 | 50.16 | 42.27 | 36.51 |
| | CWA | 40.05 | 41.01 | 31.63 | 47.90 | 42.32 | 15.34 | 56.72 | 49.97 | 42.49 | 36.51 |
| | AnyAttack | 43.00 | 43.73 | 34.77 | 50.80 | 45.24 | 17.68 | 57.54 | 53.82 | 46.17 | 40.84 |
| | X-Transfer | 42.89 | 44.94 | 34.65 | 50.70 | 43.78 | 18.85 | 55.72 | 53.16 | 46.75 | 41.12 |
| | **CAVALRY** | **43.22** | 43.42 | 34.81 | 51.60 | 45.58 | 20.39 | 56.34 | 52.58 | 45.59 | 42.19 |
| QwenVL 2.5-7B | CWA | 39.43 | 42.22 | 38.12 | 43.91 | 32.67 | 14.29 | 50.75 | 51.66 | 43.86 | 37.93 |
| | GCMA | 39.48 | 42.22 | 38.12 | 44.11 | 32.67 | 14.36 | 51.26 | 51.60 | 43.35 | 38.14 |
| | AnyAttack | 44.34 | 46.58 | 42.60 | 50.75 | 36.26 | 24.71 | 53.20 | 53.42 | 48.12 | 43.32 |
| | X-Transfer | 44.34 | 46.58 | 42.60 | 50.75 | 36.26 | 24.71 | 53.20 | 53.42 | 48.12 | 43.32 |
| | **CAVALRY** | **47.10** | 49.18 | 45.09 | 55.34 | 37.19 | 32.35 | 54.59 | 53.82 | 50.00 | 45.45 |
| Aria | GCMA | 36.74 | 32.91 | 29.68 | 42.47 | 44.64 | 15.71 | 53.96 | 48.27 | 31.21 | 34.38 |
| | CWA | 36.95 | 32.91 | 29.68 | 43.26 | 44.64 | 15.96 | 54.15 | 48.21 | 31.86 | 34.45 |
| | X-Transfer | 39.28 | 35.06 | 32.49 | 45.41 | 46.64 | 19.90 | 54.08 | 51.53 | 34.47 | 36.29 |
| | AnyAttack | 39.93 | 36.14 | 32.90 | 45.51 | 47.97 | 20.95 | 54.90 | 51.14 | 34.54 | 38.00 |
| | **CAVALRY** | **42.12** | 38.54 | 33.88 | 49.30 | 49.77 | 24.09 | 54.96 | 53.10 | 37.64 | 40.84 |
| Llava-Video 7B | CWA | 34.79 | 34.75 | 31.51 | 47.06 | 23.82 | 10.35 | 50.06 | 47.68 | 34.25 | 32.17 |
| | GCMA | 34.79 | 34.75 | 31.51 | 47.06 | 23.82 | 10.29 | 49.87 | 48.01 | 33.96 | 32.39 |
| | AnyAttack | 37.86 | 37.78 | 34.86 | 50.30 | 26.28 | 14.42 | 51.70 | 49.45 | 38.15 | 36.36 |
| | X-Transfer | 38.33 | 38.67 | 35.34 | 50.60 | 26.48 | 14.36 | 53.14 | 49.90 | 38.22 | 36.72 |
| | **CAVALRY** | **40.29** | 39.87 | 36.81 | 52.59 | 30.01 | 18.92 | 53.14 | 49.25 | 41.04 | 39.91 |
| *Commercial MLLMs* | | | | | | | | | | | |
| Gemini 2.0-Flash | CWA | 39.68 | 30.82 | 29.60 | 37.77 | 68.00 | 14.11 | 56.22 | 44.55 | 43.42 | 41.48 |
| | GCMA | 39.72 | 31.01 | 29.47 | 37.57 | 68.46 | 14.29 | 56.16 | 44.81 | 43.21 | 41.48 |
| | X-Transfer | 40.58 | 32.85 | 30.41 | 39.87 | 66.27 | 15.53 | 57.98 | 43.64 | 42.85 | 44.25 |
| | AnyAttack | 40.89 | 32.91 | 30.74 | 40.17 | 66.80 | 15.90 | 58.42 | 44.36 | 43.71 | 43.32 |
| | **CAVALRY** | **42.11** | 33.23 | 31.63 | 41.92 | 68.80 | 19.10 | 59.11 | 46.58 | 43.35 | 43.32 |
| GPT 4.1-Mini | CWA | 31.72 | 41.39 | 29.76 | 34.13 | 21.56 | 5.61 | 46.48 | 44.81 | 40.10 | 22.66 |
| | GCMA | 31.92 | 40.82 | 29.56 | 35.48 | 21.69 | 5.67 | 46.98 | 44.49 | 40.68 | 22.87 |
| | AnyAttack | 32.92 | 42.28 | 30.70 | 36.13 | 22.42 | 8.69 | 46.73 | 45.60 | 38.73 | 25.71 |
| | X-Transfer | 33.87 | 42.03 | 31.19 | 36.63 | 26.01 | 9.30 | 46.36 | 48.14 | 38.01 | 28.48 |
| | **CAVALRY** | **35.62** | 44.81 | 33.47 | 38.92 | 25.08 | 10.78 | 48.30 | 48.40 | 42.77 | 28.98 |

gap ranges from a 22.7% higher SRR on QwenVL2.5-7B to a 64.8% higher SRR on Aria. This strong performance on static images validates that our approach successfully targets the fundamental cross-modal integration vulnerabilities shared across multimodal systems, confirming the universality of our framework beyond the most challenging spatiotemporal scenarios.

## 5.4 FURTHER ANALYSIS

**Attack Mechanism Analysis**   To better understand how CAVALRY works in practice, we present a qualitative example in Figure 2. This example demonstrates three distinct scenarios: (1) *Incorrect object recognition*, shown in A1 where a Mercedes-Benz is misidentified as a Fiat; (2) *Incorrect reasoning despite accurate visual perception*, evident in A2 and A3 where the model correctly perceives visual elements but provides contextually incorrect answers; and (3) *Attack failure when answers rely on stored knowledge*, demonstrated in A4 where the model correctly describes non-Newtonian fluids despite the adversarial perturbation. These observations suggest that CAVALRY primarily disrupts the connection between visual input and language generation. When answers depend heavily on visual evidence (A1-A3), the attack successfully breaks this connection. However, when questions can be answered using knowledge already stored in the model (A4), the attack is less effective. This analysis helps us understand both when MLLMs are vulnerable to attack and when they show natural resistance.

**Efficiency Analysis**   To evaluate the computational efficiency of our method, we measure the processing time as a function of the number of input frames. The results, presented in Figure 4, demonstrate a strong linear relationship between processing time and frame count, achieving a coefficient of determination of $R^2 = 0.919$. This confirms that our method operates with a stable

Table 3: Performance comparison on the MME image benchmark, measured by Score Reduction Rate (SRR %). Higher SRR indicates a more successful attack. Best results are in **bold**.

| Model | Attack | Overall Score | Perception | | | | | | | | | | Reasoning | | | |
|---|---|---|---|---|---|---|---|---|---|---|---|---|---|---|---|---|
| | | | OCR | Art. | Celeb. | Color | Count | Exist. | Land. | Posi. | Posters | Scene | Code. | Comm. | Num. | Text. |
| *Open-Source MLLMs* | | | | | | | | | | | | | | | | |
| InternVL 2.5-8B | CWA | 4.31 | 2.78 | 4.85 | 4.20 | 3.63 | 3.81 | 5.13 | 3.51 | 4.04 | 2.83 | 3.05 | 6.25 | 3.47 | 6.90 | 3.70 |
| | AnyAttack | 3.22 | 1.39 | 2.50 | 3.22 | 2.06 | 1.90 | 2.56 | 2.05 | 2.02 | 1.80 | 1.53 | 3.12 | 2.01 | 3.45 | 1.23 |
| | X-Transfer | 4.83 | 1.39 | 3.84 | 2.15 | 1.06 | 2.89 | 2.56 | 2.92 | 3.06 | 2.23 | 2.75 | 1.56 | 0.97 | 1.72 | 2.47 |
| | **CAVALRY** | **6.34** | 5.56 | 7.07 | 5.28 | 5.19 | 5.77 | 7.69 | 4.97 | 6.13 | 3.85 | 4.58 | 7.81 | 5.00 | 8.62 | 4.94 |
| MiniCPM o-2.6 | CWA | 6.27 | 2.51 | 18.23 | 26.40 | 17.68 | 13.45 | 7.50 | 1.62 | 2.30 | 0.56 | 6.53 | -5.77 | 8.17 | -16.07 | 2.70 |
| | AnyAttack | 4.11 | 0.00 | 16.72 | 25.08 | 15.93 | 11.54 | 5.00 | -0.12 | 0.00 | -0.73 | 4.97 | -11.54 | 6.23 | -21.43 | 0.00 |
| | X-Transfer | 6.53 | 3.39 | 17.00 | 26.01 | 16.78 | 12.46 | 5.00 | 0.75 | 1.12 | -0.17 | 5.91 | -11.54 | 5.30 | -19.64 | 1.35 |
| | **CAVALRY** | **9.47** | 5.08 | 19.93 | 27.79 | 19.44 | 15.35 | 10.00 | 3.18 | 4.61 | 1.34 | 8.08 | -1.92 | 10.05 | -10.71 | 4.05 |
| QwenVL 2.5-7B | CWA | 3.62 | 2.67 | 4.14 | 4.79 | 3.60 | 4.14 | 5.26 | 3.53 | 4.45 | 3.34 | 3.45 | 5.97 | 3.85 | 6.56 | 3.00 |
| | AnyAttack | 3.58 | 1.33 | 2.15 | 3.35 | 1.80 | 2.10 | 2.63 | 2.06 | 2.23 | 2.19 | 1.88 | 2.99 | 2.23 | 3.28 | 1.33 |
| | X-Transfer | 4.66 | 2.67 | 3.35 | 1.92 | 2.73 | 3.10 | 2.63 | 2.76 | 3.37 | 2.77 | 3.14 | 2.99 | 1.62 | 4.92 | 2.67 |
| | **CAVALRY** | **5.71** | 4.00 | 6.53 | 5.36 | 5.46 | 6.18 | 7.89 | 4.82 | 6.74 | 4.50 | 5.02 | 7.46 | 5.46 | 8.20 | 2.67 |
| Aria | CWA | 4.56 | 5.88 | 4.18 | 8.01 | 10.81 | 2.97 | 7.89 | 0.00 | 9.57 | 0.95 | -4.27 | 8.51 | 0.00 | 4.26 | 4.05 |
| | AnyAttack | 4.92 | 4.41 | 7.67 | 21.34 | 17.14 | 6.90 | 5.26 | 3.66 | -1.02 | 0.17 | -1.28 | -4.26 | 3.06 | -2.13 | 4.05 |
| | X-Transfer | 3.95 | 4.41 | 6.76 | 22.62 | 17.14 | 4.93 | 2.63 | 4.84 | -1.02 | 1.90 | 0.64 | -17.02 | 0.00 | 2.13 | 0.00 |
| | **CAVALRY** | **8.11** | 8.82 | 10.80 | 26.81 | 21.62 | 2.97 | 7.89 | 7.49 | 6.38 | 0.78 | -3.19 | 14.89 | 3.65 | 8.51 | 0.00 |
| LLaVA-Video 7B | CWA | 4.35 | 8.82 | 4.31 | 4.20 | 4.27 | 3.92 | 5.13 | 4.28 | 4.57 | 3.17 | 3.53 | 6.82 | 4.16 | 7.14 | 8.33 |
| | AnyAttack | 3.04 | 5.88 | 2.90 | 2.43 | 2.17 | 1.96 | 2.56 | 2.63 | 2.32 | 1.76 | 1.47 | 4.55 | 2.04 | 4.76 | 5.56 |
| | X-Transfer | 3.93 | 2.94 | 1.73 | 1.68 | 3.19 | 2.97 | 2.56 | 3.75 | 3.41 | 2.99 | 2.35 | 2.27 | 1.02 | 2.38 | 2.78 |
| | **CAVALRY** | **6.77** | 11.76 | 5.88 | 5.63 | 6.38 | 5.94 | 7.69 | 6.05 | 6.82 | 4.40 | 5.59 | 9.09 | 6.20 | 9.52 | 11.11 |
| *Commercial MLLMs* | | | | | | | | | | | | | | | | |
| Gemini 2.0-Flash | CWA | 3.51 | -8.11 | -0.20 | 23.19 | 5.26 | 13.45 | 7.50 | 0.41 | -4.41 | -2.21 | -4.55 | 0.00 | 8.70 | 8.45 | 0.00 |
| | AnyAttack | 5.96 | 0.00 | 6.00 | 34.38 | 13.16 | 14.43 | 2.50 | -0.18 | -8.83 | -1.49 | -2.28 | 4.05 | 15.22 | -8.45 | 11.69 |
| | **X-Transfer** | **7.05** | 0.00 | 11.67 | 34.38 | 13.16 | 19.22 | 2.50 | 5.44 | -10.33 | 2.10 | -0.46 | 0.00 | 15.22 | -4.23 | 7.80 |
| | CAVALRY | 6.31 | -4.05 | 7.47 | 36.00 | 23.68 | 11.54 | 2.50 | 4.44 | -19.15 | -0.55 | -4.88 | 0.00 | 12.17 | 0.00 | 11.69 |
| GPT 4.1-Mini | CWA | 5.72 | 0.00 | 10.45 | 18.68 | -5.46 | 9.82 | 7.89 | 2.59 | 4.45 | 1.28 | 2.50 | 18.46 | 9.41 | 9.68 | 5.36 |
| | AnyAttack | 7.94 | -4.05 | 14.57 | 21.07 | 0.00 | 15.71 | 5.26 | 17.41 | 14.57 | 1.04 | -0.71 | 13.85 | 10.66 | 9.68 | 5.36 |
| | X-Transfer | 7.60 | -4.05 | 18.93 | 21.07 | 7.26 | 2.94 | 2.63 | 20.81 | 6.74 | 0.24 | 4.63 | 23.08 | 11.91 | 0.00 | 5.36 |
| | **CAVALRY** | **10.49** | 0.00 | 17.45 | 26.00 | 20.90 | 9.82 | -2.63 | 23.08 | 17.94 | -0.79 | 2.35 | 13.85 | 18.34 | 9.68 | 5.36 |

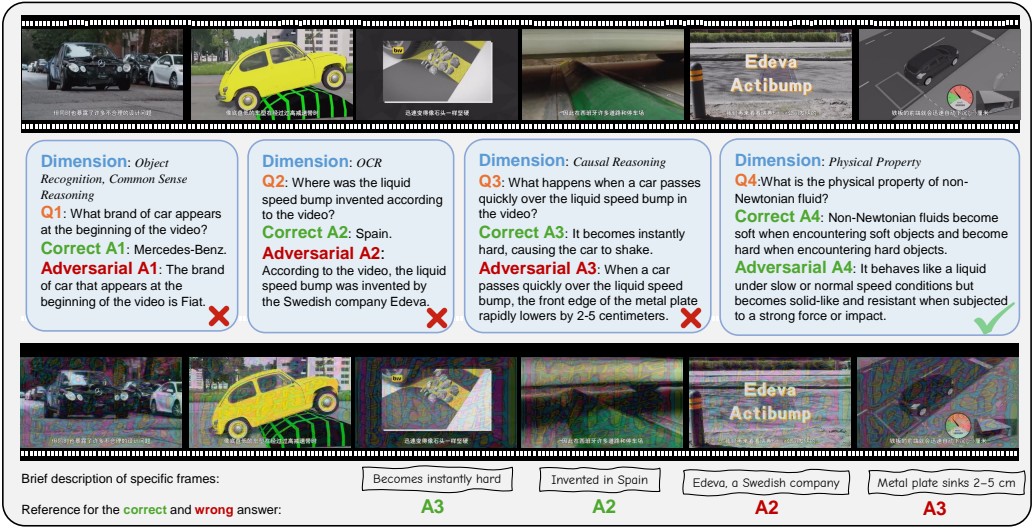

Figure 2: Visualization of attack effects on video understanding. Original video frames (top) and their adversarially perturbed versions (bottom). The attack succeeds for A1, A2 and A3, but fails for A4. Green/red text indicates correct/incorrect answers.

and predictable linear time complexity. The linear regression analysis reveals a processing speed of approximately 65 frames per second (a slope of 15.4 ms/frame). This high throughput underscores the practical applicability of our approach for real-world scenarios.

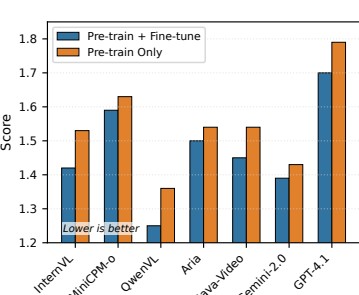

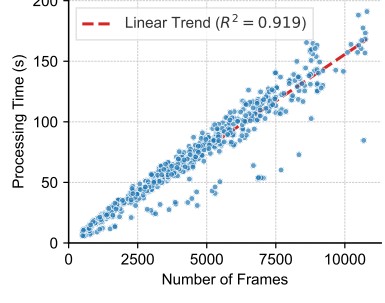

| Method | NFC |
|---|---|
| CWA | N/A |
| X-Transfer | N/A |
| AnyAttack | -4.07 |
| GCMA | -5.99 |
| Uniform Noise | -0.53 |
| **CAVALRY** | **0.84** |

Table 4: Temporal consistency comparison using NFC. Higher values indicate better temporal coherence.

Figure 3: Performance comparison between our full pipeline and using pre-training only.

Figure 4: Efficiency of our method. Processing time as a function of the number of video frames. Each blue dot represents a video sample.

**Temporal Consistency Analysis** We evaluate the temporal consistency of perturbations using the Normalized Flow Consistency (NFC) metric, which measures how well perturbations adhere to the video's motion patterns. Mathematically, NFC is defined as $\text{NFC} = 1 - \frac{\|W(\delta_{t-1}) - \delta_t\|_2}{\|\delta_t\|_2}$, where $W(\cdot)$ is the warping operation based on the optical flow between consecutive frames. NFC ranges from $(-\infty, 1]$, with higher values indicating better consistency. Table 4 presents the comparison results. Our method achieves a positive NFC score, demonstrating its ability to generate perturbations that coherently follow the temporal dynamics of the video. In contrast, all baseline methods produce negative NFC values. These results validate that our approach successfully produces consistent perturbations across temporal sequences.

## 6 REAL-WORLD THREAT ANALYSIS: BYPASSING SAFETY GUARDRAILS

To demonstrate the practical risks of CAVALRY in deployment scenarios, we conduct a case study on *Content Moderation Evasion*. We investigate two aspects: (1) the capability to bypass commercial safety guardrails, and (2) robustness against proactive defense mechanisms. We utilize the XD-Violence dataset (Wu et al., 2020), specifically a set of 500 verified real-world violent videos (e.g., abuse, explosions). We target Gemini 2.5 Flash, a state-of-the-art commercial MLLM with strong safety alignment, prompting it to act as a moderator: *"Does this video contain violent content?"* We report ASR, defined as the percentage of violent videos incorrectly classified as "Safe".

As shown in Table 5, Gemini 2.5 is highly effective on clean data, with only a 13.4% miss rate. However, CAVALRY drastically compromises this safety alignment. Under our default setting of JPEG compression (Q=75), simulating standard web transmission, the attack achieves a 54.2% ASR. To stress-test robustness against active sanitization, we evaluate four additional input transformation defenses: Stronger JPEG (Q=50), Bit-Depth Reduction (5-bit), Gaussian Blur ($\sigma = 1.0$), and Median Filtering. The attack demonstrates remarkable resilience, maintaining ASRs between 45.1% and 52.1%—ap-

Table 5: ASR on bypassing violence detection in Gemini 2.5. We evaluate robustness against five common defenses. JPEG (Q=75) is our default setting simulating web compression.

| Scenario / Defense | ASR (%) |
|---|---|
| Clean Input (Baseline) | 13.4 |
| CAVALRY (Default, JPEG=75) | 54.2 |
| + Stronger JPEG (Q=50) | 52.1 |
| + Bit-Depth Reduction (5-bit) | 50.2 |
| + Gaussian Blur ($\sigma = 1.0$) | 50.0 |
| + Median Filtering ($3 \times 3$) | 45.1 |

proximately four times the baseline miss rate. These results confirm that our perturbations are spatiotemporally coherent and robust to standard defensive measures.

These findings highlight systemic vulnerabilities in automated moderation, which is indispensable for modern platforms. The ability to sever the visual-linguistic link for violence suggests similar evasion risks for other prohibited categories, such as pornography or terrorist propaganda. The persistence of attack effectiveness under robust defenses underscores the urgent need for next-generation security measures beyond simple input transformations.

## ETHICAL CONSIDERATIONS

The development of adversarial attacks on MLLMs carries a significant responsibility. We acknowledge the dual-use potential of the CAVALRY framework. While its primary purpose is to serve as a research tool for identifying and helping to fix vulnerabilities in MLLMs (i.e., for defensive purposes), it could potentially be misused by malicious actors to circumvent safety filters and propagate harmful content (e.g., violence, hate speech, disinformation).

To mitigate these risks and promote responsible use, we will adhere to the following principles: **(i)** The CAVALRY framework and pre-trained models are released for academic research and responsible vulnerability disclosure purposes only. They are intended to be used by security researchers and model developers to build more robust systems. **(ii)** We will release the model weights under a responsible use license, requiring users to agree not to use the technology to generate or disseminate harmful, deceptive, or illegal content. **(iii)** We encourage MLLM developers to utilize our framework and findings as a red-teaming tool to strengthen their safety alignments and build defenses against such generative-disruption attacks.

This work's contribution to the broader AI safety landscape is to provide a clear, reproducible methodology for stress-testing the crucial vision-language grounding in MLLMs, ultimately fostering the development of more secure and reliable AI.

## REPRODUCIBILITY STATEMENT

We are committed to full reproducibility and will publicly release our official source code, pre-trained generator weights, and evaluation scripts. All experiments were conducted based on the configurations described below.

**(i) Training Configuration**: The generator's training process is detailed in our Implementation Details section. The most computationally intensive stage is the pre-training, which was conducted for 1.4 million steps on the LAION-400M dataset utilizing 8 NVIDIA A800 80GB GPUs. The subsequent fine-tuning stages are substantially less demanding. By releasing the pre-trained weights, we enable other researchers to bypass this intensive step and focus on fine-tuning or inference.

**(ii) Inference Configuration**: As stated in our implementation details, the inference process with the trained generator is highly efficient. The framework is capable of generating perturbations for multi-minute videos within seconds on a single GPU. This efficiency makes our method practical for large-scale MLLM security auditing and robustness evaluation.

**(iii) Evaluation Configuration**: The evaluation of adversarial examples involves two main types of resources. For commercial models such as GPT-4.1 and Gemini 2.0, reproducibility relies on access to their respective APIs, which constitutes the primary financial cost. For the open-source models evaluated in our work (e.g., InternVL-2.5 8B, QwenVL-2.5 7B), reproducibility requires hardware capable of hosting these models, typically a modern datacenter or high-end consumer GPU.

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

## A  CONCLUSION

We presented CAVALRY, a transferable generator framework for adversarial attacks on multimodal systems. Our approach introduced two key innovations: (1) a generative likelihood divergence maximization mechanism targeting the interface between visual perception and language generation, and (2) a progressive multi-stage training strategy combining large-scale foundation pretraining with specialized fine-tuning for spatiotemporal coherence. Extensive experiments demonstrated that our method consistently outperforms existing approaches across diverse multimodal architectures, from the most challenging spatiotemporal scenarios to static image understanding. This work represents a significant step toward understanding and addressing the fundamental vulnerabilities of cross-modal integration in multimodal systems.

## LLM USAGE

We utilized LLMs as an assistive tool in the preparation of this work. Specifically, LLMs were employed to: (i) refine phrasing, improve the clarity and structure of the manuscript, and act as a reviewer by providing feedback; and (ii) assist in generating code and debugging.

## B  EXPERIMENTS

### B.1  EVALUATION DETAILS

We used the VLMEVALKIT TOOLKIT[2] to conduct evaluations on both the MMBench-Video and MME benchmarks. In line with the toolkit's methodology, LLMs served as automatic evaluators to assess the semantic alignment between model responses and ground-truth answers. While the MME benchmark defaults to GPT-4o-mini and MMBench-Video uses GPT-4-turbo as the judge, we standardized our evaluation process for consistency by using **GPT-4o-mini** for both benchmarks. We maintained the default generation parameters for the judge (e.g., a temperature of 0) to ensure deterministic and comparable scores.

### B.2  BASELINE METHOD DETAILS

We selected state-of-the-art approaches as baselines based on our analysis of attack categories.

- **GCMA** (Chen et al., 2023) (ACM MM 2023): An efficient generator-based method designed for single-modality vision models.

- **CWA** (Chen et al., 2024a) (ICLR 2024): An image attack extensible to video domains. Due to its iterative generation process and high computational demands, we adapted it by applying identical perturbations to all frames within each one-second segment of the video.

- **AnyAttack** (Zhang et al., 2025b) (CVPR 2025): A generator-based multimodal attack. Since the officially released weights were trained for targeted attacks, we adapted it for our untargeted setting by deploying it with randomized targets to align with our misclassification objective.

- **X-Transfer** (Huang et al., 2025) (ICML 2025): A Universal Adversarial Perturbation (UAP) approach for multimodal models. We applied their officially released perturbation pattern uniformly across all frames of the video inputs.

### B.3  CAVALRY TRAINING AND INFERENCE DETAILS

**Surrogate and Auxiliary Models**  We employed InternVL-2.5-1B as our white-box surrogate model $\mathcal{M}$. For the auxiliary feature loss $\mathcal{L}_{aux}$, we utilized an adversarially trained ResNet-50 on ImageNet (Russakovsky et al., 2015).

---

[2] https://github.com/open-compass/VLMEvalKit

**Training Configuration**    Our generator training process was conducted on 8 NVIDIA A800 80GB GPUs. We used the AdamW optimizer with a learning rate of $1 \times 10^{-4}$. The training was divided into two phases:

1. **Pre-training**: The generator was trained for 1,400,000 steps on the LAION-400M dataset with a batch size of 8 per GPU.

2. **Instruction and video Fine-tuning**: We continued training for 20 epochs on the LLaVA-Instruct-150K dataset, which consisted of image-only data, using a cosine learning rate annealing schedule. Finally, we fine-tuned on the Video-MME dataset. For each video, we uniformly sampled 50 batches of frames, maintaining a total batch size of 64 (8 GPUs $\times$ 8 samples).

**Inference**    Once trained, the generator $G$ can produce perturbations efficiently. For videos in the MMBench-Video benchmark with frame counts less than or equal to 300, we processed the entire sequence in a single forward pass. For longer videos, we used sequential processing with a batch size of 300 frames per pass. This strategy enables perturbation generation for multi-minute videos within seconds.

### B.4    ADDITIONAL EXPERIMENTAL RESULTS

**Fine-grained Performance Analysis**    To provide a comprehensive breakdown of our results, this section presents the fine-grained performance across all sub-categories from the MMBench-Video benchmark. The following tables report the **original benchmark scores**.

**Important Note**: For these appendix tables, unlike the SRR metric used in the main paper, a **lower score indicates a more effective attack**. The *Clean* row in each section serves as the baseline performance for comparison.

Tables 6 and 7 present the detailed category-specific results that constitute the aggregated scores reported in the main paper. Table 6 details performance across various perception-related tasks (e.g., Video Topic, Object Recognition), while Table 7 extends this analysis to complex reasoning capabilities (e.g., Causal Reasoning, Future Prediction). These tables provide a complete view of the attack's impact across all evaluated dimensions.

## C    QUANTITATIVE ANALYSIS OF VISUAL DISTORTION

While the experiments in the main text utilize a maximum perturbation budget of $\epsilon = 16/255$ to evaluate peak adversarial capability, we provide a supplementary quantitative analysis here to assess visual perceptibility across varying budgets ($\epsilon \in \{4/255, 8/255, 16/255\}$). We employ the average $L_2$ norm as the distortion metric and compare our approach with XTransfer, Any, CWA, and GCMA.

As illustrated in Figure 5, we observe a distinct linear correlation between the perturbation budget and the average $L_2$ norm across all evaluated methods. At equivalent $\epsilon$ levels, our method exhibits distortion magnitudes that are highly consistent with XTransfer and Any (e.g., clustering around $L_2 \approx 5.9$ at $\epsilon = 4/255$ and $L_2 \approx 22.5$

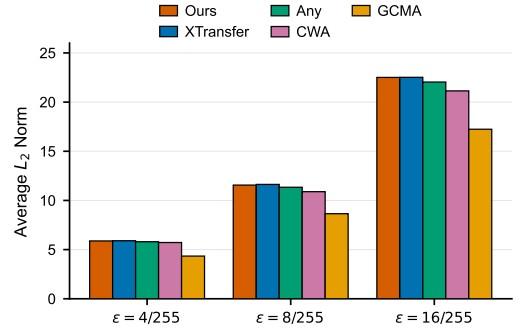

Figure 5: Quantitative comparison of visual distortion ($L_2$ Norm) across varying perturbation budgets on TempCompass dataset.

at $\epsilon = 16/255$). This alignment indicates that these approaches uniformly utilize the available budget to maximize transferability and attack potency. Although larger budgets naturally yield stronger attacks at the cost of higher distortion, our framework offers a flexible trade-off, providing a stealthy option ($\epsilon = 4/255$) with significantly reduced distortion energy alongside the high-performance setting. Notably, while GCMA demonstrates lower $L_2$ norms due to its specific optimization constraints,

Table 6: Fine-grained results for perception tasks on the MMBench-Video benchmark. All values are the original benchmark scores, where a **lower score indicates a more successful attack**. VT: Video Topic, VE: Video Emotion, VS: Video Scene, VSt: Video Style, OCR: OCR, OR: Object Recognition, AR: Attribute Recognition, ER: Event Recognition, HM: Human Motion, C: Counting, SR: Spatial Relationship, HOI: Human-object Interaction, HI: Human Interaction.

| Model | Method | VT | VE | VS | VSt | OCR | OR | AR | ER | HM | C | SR | HOI | HI |
|---|---|---|---|---|---|---|---|---|---|---|---|---|---|---|
| InternVL2.5 8B | Clean | 1.71 | 1.65 | 1.70 | 1.93 | 1.79 | 1.81 | 2.04 | 1.49 | 1.44 | 1.78 | 1.67 | 1.41 | 1.49 |
| | GCMA | 1.68 | 1.65 | 1.63 | 1.90 | 1.80 | 1.71 | 1.96 | 1.37 | 1.28 | 1.82 | 1.60 | 1.35 | 1.41 |
| | CWA | 1.69 | 1.53 | 1.71 | 1.60 | 1.77 | 1.64 | 1.90 | 1.34 | 1.24 | 1.65 | 1.42 | 1.32 | 1.46 |
| | Any-attack | 1.66 | 1.53 | 1.68 | 1.64 | 1.77 | 1.65 | 1.90 | 1.34 | 1.20 | 1.64 | 1.47 | 1.32 | 1.41 |
| | X-Transfer | 1.61 | 1.51 | 1.54 | 1.52 | 1.71 | 1.51 | 1.87 | 1.36 | 1.11 | 1.64 | 1.53 | 1.19 | 1.28 |
| | **CAVALRY** | 1.64 | 1.51 | 1.56 | 1.69 | 1.70 | 1.37 | 1.76 | 1.35 | 1.10 | 1.64 | 1.38 | 1.10 | 1.28 |
| MiniCPM-o 2.6 | Clean | 1.82 | 1.73 | 1.93 | 1.83 | 2.04 | 1.92 | 2.06 | 1.52 | 1.39 | 1.66 | 1.71 | 1.52 | 1.57 |
| | GCMA | 1.81 | 1.62 | 1.77 | 1.86 | 1.99 | 1.82 | 2.05 | 1.58 | 1.39 | 1.59 | 1.60 | 1.49 | 1.54 |
| | CWA | 1.71 | 1.65 | 1.67 | 1.50 | 1.99 | 1.69 | 1.87 | 1.40 | 1.29 | 1.54 | 1.60 | 1.37 | 1.48 |
| | Any-attack | 1.72 | 1.63 | 1.70 | 1.45 | 1.99 | 1.69 | 1.87 | 1.40 | 1.27 | 1.54 | 1.64 | 1.32 | 1.48 |
| | X-Transfer | 1.81 | 1.58 | 1.79 | 1.38 | 1.99 | 1.70 | 1.91 | 1.43 | 1.31 | 1.53 | 1.42 | 1.49 | 1.44 |
| | **CAVALRY** | 1.81 | 1.56 | 1.74 | 1.43 | 1.95 | 1.56 | 1.89 | 1.39 | 1.34 | 1.49 | 1.51 | 1.28 | 1.51 |
| Qwen2.5-VL 7B | Clean | 1.92 | 1.38 | 1.40 | 2.10 | 1.56 | 1.35 | 1.77 | 1.20 | 1.08 | 1.29 | 1.36 | 1.06 | 1.02 |
| | GCMA | 1.90 | 1.42 | 1.26 | 2.14 | 1.54 | 1.25 | 1.77 | 1.15 | 1.08 | 1.30 | 1.24 | 1.15 | 1.10 |
| | CWA | 1.83 | 1.34 | 1.02 | 2.00 | 1.51 | 1.16 | 1.68 | 1.12 | 0.98 | 1.22 | 1.36 | 0.89 | 1.05 |
| | Any-attack | 1.78 | 1.27 | 1.11 | 1.88 | 1.50 | 1.07 | 1.58 | 0.99 | 1.02 | 1.14 | 0.98 | 0.95 | 1.05 |
| | X-Transfer | 1.83 | 1.34 | 0.93 | 1.76 | 1.45 | 1.08 | 1.55 | 1.05 | 1.00 | 1.12 | 1.11 | 0.88 | 1.00 |
| | **CAVALRY** | 1.79 | 1.19 | 0.94 | 1.76 | 1.46 | 1.02 | 1.39 | 1.07 | 0.82 | 1.12 | 1.22 | 0.77 | 0.89 |
| Aria | Clean | 1.80 | 1.63 | 1.54 | 2.00 | 1.60 | 1.67 | 1.95 | 1.43 | 1.48 | 1.47 | 1.49 | 1.29 | 1.26 |
| | GCMA | 1.78 | 1.75 | 1.59 | 1.95 | 1.54 | 1.66 | 1.90 | 1.39 | 1.34 | 1.62 | 1.53 | 1.41 | 1.23 |
| | CWA | 1.69 | 1.72 | 1.44 | 1.88 | 1.53 | 1.56 | 1.88 | 1.42 | 1.39 | 1.46 | 1.47 | 1.40 | 1.28 |
| | Any-attack | 1.66 | 1.67 | 1.26 | 1.93 | 1.57 | 1.50 | 1.80 | 1.36 | 1.30 | 1.48 | 1.38 | 1.27 | 1.28 |
| | X-Transfer | 1.66 | 1.71 | 1.28 | 1.74 | 1.54 | 1.42 | 1.89 | 1.34 | 1.38 | 1.55 | 1.13 | 1.37 | 1.20 |
| | **CAVALRY** | 1.66 | 1.65 | 1.35 | 1.79 | 1.52 | 1.42 | 1.86 | 1.39 | 1.39 | 1.43 | 1.07 | 1.23 | 1.26 |
| LLaVA-Video 7b | Clean | 1.58 | 1.77 | 1.79 | 2.00 | 1.74 | 1.76 | 1.90 | 1.60 | 1.33 | 1.64 | 1.40 | 1.50 | 1.34 |
| | GCMA | 1.55 | 1.75 | 1.76 | 1.67 | 1.74 | 1.76 | 1.91 | 1.58 | 1.41 | 1.67 | 1.51 | 1.45 | 1.28 |
| | CWA | 1.49 | 1.71 | 1.62 | 1.79 | 1.70 | 1.68 | 1.86 | 1.48 | 1.37 | 1.59 | 1.38 | 1.41 | 1.39 |
| | Any-attack | 1.44 | 1.67 | 1.55 | 1.76 | 1.70 | 1.63 | 1.79 | 1.40 | 1.25 | 1.67 | 1.44 | 1.43 | 1.33 |
| | X-Transfer | 1.32 | 1.70 | 1.55 | 1.57 | 1.69 | 1.60 | 1.83 | 1.43 | 1.34 | 1.58 | 1.47 | 1.41 | 1.33 |
| | **CAVALRY** | 1.41 | 1.61 | 1.60 | 1.69 | 1.68 | 1.49 | 1.78 | 1.39 | 1.26 | 1.62 | 1.47 | 1.31 | 1.31 |
| Gemini-2.0 Flash | Clean | 2.36 | 0.50 | 1.71 | 2.00 | 1.69 | 1.71 | 2.40 | 1.50 | 1.67 | 1.68 | 1.11 | 1.17 | 1.31 |
| | GCMA | 2.43 | 0.67 | 1.43 | 2.14 | 1.47 | 1.39 | 2.33 | 1.29 | 1.83 | 1.45 | 1.11 | 1.00 | 1.31 |
| | CWA | 2.14 | 0.67 | 1.57 | 2.14 | 1.44 | 1.52 | 2.40 | 1.57 | 1.33 | 1.45 | 1.44 | 1.25 | 1.15 |
| | Any-attack | 2.36 | 0.50 | 1.43 | 1.43 | 1.29 | 1.35 | 2.20 | 1.57 | 1.50 | 1.73 | 0.44 | 1.00 | 1.00 |
| | X-Transfer | 2.07 | 0.33 | 1.71 | 1.86 | 1.49 | 1.45 | 2.13 | 1.50 | 1.00 | 2.00 | 0.67 | 1.17 | 1.08 |
| | **CAVALRY** | 2.14 | 0.33 | 1.43 | 1.57 | 1.31 | 1.29 | 2.13 | 1.29 | 1.17 | 1.77 | 0.89 | 1.17 | 0.77 |
| GPT4.1 Mini | Clean | 2.64 | 1.33 | 2.43 | 2.71 | 2.16 | 1.52 | 2.47 | 2.00 | 1.50 | 2.00 | 2.11 | 1.17 | 1.38 |
| | GCMA | 2.50 | 1.33 | 2.00 | 2.71 | 1.87 | 1.48 | 2.40 | 1.57 | 1.67 | 1.82 | 2.22 | 1.00 | 1.23 |
| | CWA | 2.43 | 1.33 | 1.57 | 2.57 | 2.02 | 1.35 | 2.47 | 1.57 | 1.17 | 1.50 | 2.22 | 1.25 | 1.54 |
| | Any-attack | 2.29 | 1.17 | 1.29 | 2.71 | 1.80 | 1.48 | 2.47 | 1.79 | 1.00 | 1.50 | 2.00 | 1.17 | 1.62 |
| | X-Transfer | 2.14 | 1.17 | 1.86 | 2.71 | 1.73 | 1.39 | 2.13 | 1.71 | 1.17 | 1.64 | 2.22 | 1.17 | 1.62 |
| | **CAVALRY** | 2.14 | 0.83 | 1.57 | 2.57 | 1.73 | 1.48 | 2.27 | 1.50 | 1.17 | 1.77 | 2.11 | 1.42 | 1.69 |

Table 7: Fine-grained results for reasoning tasks on the MMBench-Video benchmark. All values are the original benchmark scores, where a **lower score indicates a more successful attack**. Hal: Hallucination, SIT: Structuralized Image-Text Understanding, MC: Mathematical Calculation, PP: Physical Property, FR: Function Reasoning, IR: Identity Reasoning, NR: Natural Relation, PR: Physical Relation, SR: Social Relation, CSR: Common Sense Reasoning, CFR: Counterfactual Reasoning, CR: Causal Reasoning, FP: Future Prediction.

| Model | Method | Hal | SIT | MC | PP | FR | IR | NR | PR | SR | CSR | CFR | CR | FP |
|---|---|---|---|---|---|---|---|---|---|---|---|---|---|---|
| InternVL2.5 8B | Clean | 0.56 | 1.12 | 0.71 | 1.48 | 1.45 | 1.87 | 1.56 | 1.65 | 2.00 | 1.48 | 1.52 | 1.41 | 1.23 |
| | GCMA | 0.65 | 1.09 | 0.73 | 1.41 | 1.38 | 1.91 | 1.48 | 1.65 | 2.02 | 1.49 | 1.43 | 1.37 | 1.28 |
| | CWA | 0.55 | 1.04 | 0.89 | 1.45 | 1.36 | 1.92 | 1.41 | 1.41 | 1.89 | 1.46 | 1.44 | 1.34 | 1.26 |
| | Any-attack | 0.53 | 1.07 | 0.89 | 1.44 | 1.36 | 1.87 | 1.44 | 1.47 | 1.87 | 1.47 | 1.38 | 1.34 | 1.21 |
| | X-Transfer | 0.39 | 1.16 | 0.87 | 1.39 | 1.33 | 1.62 | 1.44 | 1.41 | 1.89 | 1.32 | 1.43 | 1.24 | 1.15 |
| | **CAVALRY** | 0.44 | 1.21 | 0.80 | 1.41 | 1.16 | 1.49 | 1.26 | 1.31 | 1.67 | 1.38 | 1.35 | 1.34 | 1.23 |
| MiniCPM-o 2.6 | Clean | 0.84 | 1.65 | 1.18 | 1.67 | 1.53 | 1.72 | 1.74 | 1.37 | 1.85 | 1.78 | 1.45 | 1.56 | 1.53 |
| | GCMA | 0.76 | 1.71 | 1.16 | 1.54 | 1.55 | 1.60 | 1.89 | 1.55 | 1.83 | 1.90 | 1.48 | 1.57 | 1.49 |
| | CWA | 0.73 | 1.50 | 1.27 | 1.52 | 1.42 | 1.62 | 1.78 | 1.51 | 1.69 | 1.85 | 1.43 | 1.57 | 1.51 |
| | Any-attack | 0.74 | 1.50 | 1.32 | 1.54 | 1.42 | 1.60 | 1.78 | 1.53 | 1.73 | 1.85 | 1.40 | 1.56 | 1.53 |
| | X-Transfer | 0.65 | 1.50 | 1.27 | 1.44 | 1.38 | 1.64 | 1.74 | 1.43 | 1.57 | 1.95 | 1.52 | 1.52 | 1.45 |
| | **CAVALRY** | 0.73 | 1.59 | 1.16 | 1.48 | 1.40 | 1.45 | 1.74 | 1.49 | 1.67 | 1.81 | 1.62 | 1.62 | 1.56 |
| Qwen2.5-VL 7B | Clean | 1.13 | 1.60 | 1.60 | 1.80 | 1.36 | 1.64 | 1.85 | 1.59 | 1.35 | 1.48 | 1.25 | 1.17 | 1.13 |
| | GCMA | 1.13 | 1.65 | 1.69 | 1.61 | 1.81 | 1.81 | 1.33 | 1.33 | 1.59 | 1.59 | 1.30 | 1.12 | 1.09 |
| | CWA | 1.15 | 1.41 | 1.78 | 1.56 | 1.42 | 1.58 | 1.81 | 1.41 | 1.31 | 1.53 | 1.27 | 1.02 | 0.98 |
| | Any-attack | 1.08 | 1.46 | 1.76 | 1.48 | 1.36 | 1.49 | 1.74 | 1.37 | 1.07 | 1.60 | 1.25 | 1.01 | 0.99 |
| | X-Transfer | 1.11 | 1.59 | 1.96 | 1.35 | 1.40 | 1.43 | 1.81 | 1.24 | 1.02 | 1.49 | 1.12 | 1.06 | 1.09 |
| | **CAVALRY** | 1.18 | 1.44 | 1.89 | 1.46 | 1.27 | 1.32 | 1.67 | 1.27 | 0.83 | 1.49 | 1.25 | 0.99 | 1.04 |
| Aria | Clean | 0.87 | 1.72 | 1.44 | 1.80 | 1.78 | 1.79 | 1.96 | 1.43 | 1.76 | 1.78 | 1.43 | 1.51 | 1.45 |
| | GCMA | 0.94 | 1.75 | 1.40 | 1.63 | 1.75 | 1.98 | 1.74 | 1.59 | 1.81 | 1.75 | 1.38 | 1.43 | 1.38 |
| | CWA | 1.00 | 1.71 | 1.51 | 1.74 | 1.71 | 1.94 | 1.70 | 1.45 | 1.80 | 1.75 | 1.45 | 1.45 | 1.47 |
| | Any-attack | 0.66 | 1.68 | 1.47 | 1.67 | 1.69 | 1.75 | 1.70 | 1.39 | 1.63 | 1.72 | 1.35 | 1.41 | 1.34 |
| | X-Transfer | 0.65 | 1.76 | 1.36 | 1.56 | 1.65 | 1.74 | 1.78 | 1.31 | 1.69 | 1.83 | 1.45 | 1.32 | 1.51 |
| | **CAVALRY** | 0.74 | 1.65 | 1.53 | 1.57 | 1.49 | 1.79 | 1.63 | 1.47 | 1.65 | 1.73 | 1.43 | 1.35 | 1.34 |
| LLaVA-Video 7b | Clean | 0.23 | 1.41 | 1.07 | 1.67 | 1.69 | 1.64 | 1.59 | 1.41 | 1.35 | 1.72 | 1.18 | 1.47 | 1.36 |
| | GCMA | 0.16 | 1.46 | 1.02 | 1.57 | 1.62 | 1.60 | 1.67 | 1.39 | 1.31 | 1.59 | 1.27 | 1.45 | 1.47 |
| | CWA | 0.18 | 1.37 | 1.22 | 1.59 | 1.47 | 1.83 | 1.63 | 1.39 | 1.35 | 1.64 | 1.18 | 1.30 | 1.47 |
| | Any-attack | 0.18 | 1.38 | 1.18 | 1.57 | 1.56 | 1.58 | 1.52 | 1.33 | 1.20 | 1.56 | 1.18 | 1.25 | 1.34 |
| | X-Transfer | 0.19 | 1.41 | 1.13 | 1.50 | 1.49 | 1.49 | 1.78 | 1.24 | 1.19 | 1.62 | 1.20 | 1.20 | 1.19 |
| | **CAVALRY** | 0.18 | 1.37 | 1.13 | 1.48 | 1.55 | 1.55 | 1.56 | 1.33 | 1.20 | 1.46 | 1.18 | 1.18 | 1.36 |
| Gemini-2.0 Flash | Clean | 1.25 | 1.71 | 0.33 | 1.82 | 1.71 | 1.90 | 1.00 | 1.14 | 1.22 | 1.25 | 2.20 | 1.17 | 1.14 |
| | GCMA | 0.75 | 1.57 | 0.00 | 1.91 | 1.57 | 1.80 | 1.00 | 1.29 | 1.33 | 1.33 | 2.00 | 1.17 | 1.00 |
| | CWA | 0.75 | 1.57 | 1.33 | 1.64 | 1.79 | 1.60 | 0.67 | 1.71 | 0.67 | 1.42 | 1.60 | 1.28 | 0.71 |
| | Any-attack | 1.50 | 1.57 | 2.00 | 1.64 | 1.36 | 1.60 | 0.67 | 1.57 | 0.67 | 1.83 | 1.80 | 1.17 | 1.00 |
| | X-Transfer | 1.00 | 1.57 | 0.00 | 1.27 | 1.43 | 1.70 | 1.00 | 1.57 | 0.78 | 1.50 | 2.20 | 1.22 | 1.14 |
| | **CAVALRY** | 0.50 | 1.57 | 2.00 | 1.45 | 1.50 | 1.30 | 1.00 | 1.57 | 0.78 | 1.58 | 1.80 | 1.11 | 0.71 |
| GPT4.1 Mini | Clean | 0.25 | 2.14 | 0.67 | 2.18 | 1.71 | 1.80 | 2.00 | 2.29 | 1.78 | 2.25 | 1.80 | 1.72 | 1.29 |
| | GCMA | 0.25 | 2.00 | 1.00 | 2.00 | 1.86 | 1.90 | 2.00 | 2.29 | 1.89 | 2.17 | 1.60 | 1.56 | 1.14 |
| | CWA | 0.25 | 2.00 | 0.67 | 2.09 | 1.64 | 1.80 | 2.00 | 2.14 | 1.67 | 2.08 | 1.20 | 1.61 | 1.14 |
| | Any-attack | 0.50 | 2.14 | 1.00 | 2.00 | 1.71 | 1.70 | 2.00 | 2.14 | 1.67 | 1.92 | 1.60 | 1.56 | 1.43 |
| | X-Transfer | 0.25 | 2.29 | 0.67 | 2.00 | 1.71 | 1.80 | 2.00 | 2.29 | 1.44 | 1.83 | 2.00 | 1.61 | 1.14 |
| | **CAVALRY** | 0.25 | 1.71 | 0.67 | 2.09 | 1.50 | 1.40 | 2.00 | 2.29 | 1.56 | 2.08 | 1.40 | 1.56 | 0.71 |

Table 8: Reproducibility analysis on GPT-4.1-Mini over 3 independent end-to-end runs. The extremely low standard deviation ($\pm 0.13$) confirms the stability of our attack.

| Run ID | Overall | MC | Yes/No | Cap-M | Cap | Action | Direct. | Speed | Order | Att |
|---|---|---|---|---|---|---|---|---|---|---|
| Run #1 | 35.62 | 44.81 | 33.47 | 25.08 | 38.92 | 10.78 | 48.30 | 48.40 | 42.77 | 28.98 |
| Run #2 | 35.37 | 44.81 | 33.06 | 24.55 | 38.87 | 10.78 | 48.24 | 47.36 | 41.76 | 29.83 |
| Run #3 | 35.45 | 44.75 | 33.14 | 25.48 | 38.42 | 11.15 | 47.99 | 46.71 | 42.12 | 30.47 |
| **Mean $\pm$ Std** | **35.48**$_{\pm 0.13}$ | 44.79$_{\pm 0.03}$ | 33.22$_{\pm 0.22}$ | 25.04$_{\pm 0.47}$ | 38.74$_{\pm 0.27}$ | 10.90$_{\pm 0.21}$ | 48.18$_{\pm 0.16}$ | 47.49$_{\pm 0.85}$ | 42.22$_{\pm 0.51}$ | 29.76$_{\pm 0.75}$ |

this reduction comes at the expense of a substantial drop in attack success rates compared to our method, highlighting the efficiency of our approach in balancing distortion and efficacy.

## D HYPERPARAMETER SENSITIVITY ANALYSIS

To determine the optimal configuration for our loss function, we conducted a comprehensive sensitivity analysis focusing on the three key hyperparameters: $\lambda_1$, $\lambda_2$, and $\lambda_3$. We selected QwenVL-2.5 7B as the representative test model for this evaluation. Crucially, to ensure the analysis reflects the system's final capabilities, all experiments were performed using the adversarial generator from the second stage of our training pipeline (i.e., after fine-tuning for spatiotemporal coherence).

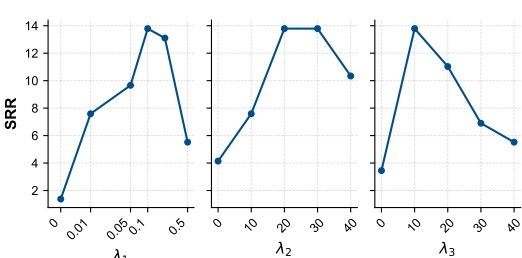

Figure 6 illustrates the impact of varying each hyperparameter while fixing the others to their default values ($\lambda_1 = 0.1, \lambda_2 = 20, \lambda_3 = 10$).

Figure 6: Hyperparameter sensitivity analysis conducted on QwenVL-2.5 7B. We report the SRR by varying $\lambda_1$ (left), $\lambda_2$ (middle), and $\lambda_3$ (right).

The results demonstrate that our method remains robust within a reasonable range of variations while exhibiting clear optimal regions. Specifically, for the semantic weight $\lambda_1$, we observe a distinct performance peak at 0.1; notably, removing this component ($\lambda_1 = 0$) leads to a collapse in attack effectiveness, underscoring the necessity of semantic guidance. Regarding the feature dimension $\lambda_2$, the attack performance improves sharply as the dimension increases from 0 to 20 and stabilizes thereafter, suggesting that $\lambda_2 = 20$ provides an efficient balance between representation capacity and computational cost. Finally, the smoothing factor $\lambda_3$ achieves its sweet spot at 10, confirming that moderate regularization is essential for maintaining the spatiotemporal consistency required to mislead video MLLMs effectively.

## E STATISTICAL SIGNIFICANCE AND REPRODUCIBILITY ANALYSIS

To strictly address concerns regarding the stochastic nature of LLM-based evaluations and ensure the statistical significance of our reported improvements, we conducted a rigorous reproducibility study. We went beyond merely repeating the LLM scoring step; instead, we re-executed the entire inference pipeline—from feeding the adversarial video frames into the MLLM to the final answer generation and scoring—three independent times.

We selected two representative commercial models, GPT-4.1-Mini and Gemini-2.0-Flash, and evaluated them on the TempCompass benchmark using our CAVALRY attack. Tables 8 and 9 report the detailed results of each independent run, along with the aggregated **Mean $\pm$ Standard Deviation**. The results demonstrate exceptional stability. The standard deviation for the Overall ASR is negligible: **0.13% for GPT-4.1-Mini** and **0.05% for Gemini-2.0-Flash**. As evidenced by the run-to-run consistency in the tables, the variations are minimal even in fine-grained categories. This confirms that the improvements reported in this paper are statistically significant.

Table 9: Reproducibility analysis on Gemini-2.0-Flash over 3 independent end-to-end runs. The results show high consistency with minimal variance ($\pm 0.05$).

| Run ID | Overall | MC | Yes/No | Cap-M | Cap | Action | Direct. | Speed | Order | Att |
|---|---|---|---|---|---|---|---|---|---|---|
| Run #1 | 42.11 | 33.23 | 31.63 | 68.80 | 41.92 | 19.10 | 59.11 | 46.58 | 43.35 | 43.32 |
| Run #2 | 42.11 | 33.16 | 31.55 | 69.06 | 41.87 | 18.98 | 59.05 | 46.38 | 43.71 | 43.39 |
| Run #3 | 42.20 | 33.04 | 31.80 | 68.80 | 42.22 | 19.22 | 59.23 | 46.77 | 43.21 | 43.47 |
| **Mean $\pm$ Std** | **42.14**$_{\pm0.05}$ | 33.14$_{\pm0.10}$ | 31.66$_{\pm0.13}$ | 68.89$_{\pm0.15}$ | 42.00$_{\pm0.18}$ | 19.10$_{\pm0.12}$ | 59.13$_{\pm0.09}$ | 46.58$_{\pm0.20}$ | 43.42$_{\pm0.26}$ | 43.39$_{\pm0.08}$ |

Table 10: Cross-validation on QwenVL-2.5-7B using three different LLM evaluators. We report the SRR %. Best results are in **bold**.

| Evaluator | Attack | Overall SRR | Perception | | | | | Reasoning | | | | | |
|---|---|---|---|---|---|---|---|---|---|---|---|---|---|
| | | | CP | FP-S | FP-C | HL | Mean | LR | AR | RR | CSR | TR | Mean |
| GPT-4o mini | GCMA | 1.4 | 2.4 | 2.8 | -3.6 | 0.0 | 2.1 | -3.8 | -1.9 | 7.7 | -7.4 | 1.7 | 0.7 |
| | CWA | 6.2 | 10.2 | 7.6 | 7.1 | -1.8 | 7.6 | 2.5 | 5.0 | 6.5 | -3.4 | 9.4 | 5.5 |
| | AnyAttack | 9.7 | 12.0 | 10.3 | 11.6 | 4.4 | 10.4 | 1.3 | 9.4 | 14.2 | -8.1 | 8.5 | 6.9 |
| | X-Transfer | 10.3 | 13.2 | 12.4 | 13.4 | 1.8 | 11.8 | -8.1 | 12.5 | 18.1 | -0.7 | 8.5 | 7.6 |
| | CAVALRY | **13.8** | 16.8 | 14.5 | 18.8 | -4.4 | **14.6** | -1.3 | 15.6 | 24.5 | -0.7 | 12.0 | **11.7** |
| Gemini 2.5-Flash | GCMA | 2.4 | 4.6 | 3.7 | -3.3 | -6.2 | 2.4 | -1.9 | -1.0 | 12.7 | 0.0 | 5.9 | 3.0 |
| | CWA | 3.6 | 4.6 | 6.2 | 0.0 | -11.3 | 3.7 | 8.7 | 5.0 | 5.8 | 1.8 | 2.2 | 4.8 |
| | AnyAttack | 9.0 | 10.2 | 9.3 | 12.4 | -14.9 | 7.9 | 6.8 | 11.6 | 15.0 | 11.4 | 11.8 | 9.6 |
| | X-Transfer | 10.2 | 10.7 | 8.7 | 14.0 | -13.9 | 10.4 | -2.5 | 13.6 | 22.0 | 18.7 | 7.4 | 12.0 |
| | CAVALRY | **13.3** | 13.2 | 14.9 | 24.0 | -16.0 | **14.0** | 0.6 | 13.1 | 23.7 | 4.8 | 12.5 | **12.0** |
| Claude 3.5-Haiku | GCMA | -1.3 | 3.0 | -0.4 | -1.0 | -10.0 | -0.4 | -6.0 | -4.4 | -4.0 | 3.7 | -3.2 | -2.2 |
| | CWA | 1.3 | 7.5 | 1.3 | -3.6 | 3.6 | 1.3 | -7.6 | 4.0 | 4.8 | 5.2 | 10.6 | 3.7 |
| | AnyAttack | 5.9 | 7.5 | 4.6 | 13.0 | -1.8 | 6.3 | -6.8 | 0.0 | 10.4 | 16.8 | 10.6 | 2.9 |
| | X-Transfer | 5.4 | 8.2 | 7.1 | 4.2 | -4.5 | 6.7 | -8.8 | 5.2 | -0.4 | 6.7 | 2.3 | 0.8 |
| | CAVALRY | **7.1** | 11.9 | 7.9 | 13.0 | -0.5 | **8.8** | -6.4 | -1.2 | 14.9 | 11.2 | 10.1 | **4.1** |

# F  ANALYSIS OF EVALUATOR BIAS

A potential concern in evaluating MLLM robustness is the reliance on a single LLM as a judge, which may introduce specific evaluator biases. To rigorously mitigate this concern and cross-validate our findings, we extended our evaluation framework beyond the default GPT-4o-mini evaluator. Specifically, we incorporated two additional state-of-the-art LLMs with distinct architectures and alignment preferences—Gemini-2.5-Flash and Claude-3.5-Haiku—to serve as independent judges. We conducted this cross-validation on the QwenVL-2.5-7B model, recalculating the SRR based on the specific scoring distribution of each evaluator.

The results demonstrate that while absolute scoring distributions vary among judges, the relative performance ranking of the attack methods remains invariant. Our method consistently achieves the highest SRR across all three evaluators, outperforming the strongest baselines. This consistency confirms that CAVALRY is an evaluator-agnostic attack, effectively disrupting the target model's semantic processing capabilities regardless of the specific judge employed.

