# OpenReview forum: "Severing the Link: A Unified Adversarial Attack on Image and Video MLLMs via Generative Disruption"
_ICLR.cc/2026/Conference — Submitted to ICLR 2026_

### Official Review · Reviewer_pStB · 2025-10-25

**Soundness:** 3
**Presentation:** 3
**Contribution:** 2
**Rating:** 4
**Confidence:** 3

**Summary:**

The paper introduces CAVALRY, a unified adversarial attack framework that disrupts the vision-language grounding in MLLMs. CAVALRY employs a two-stage training strategy to produce spatiotemporally coherent perturbations for both images and videos, achieving strong transferability across diverse MLLM architectures.

**Strengths:**

1. The motivation is clear and the task is interesting. Experimental results on seven mainstream MLLMs show that CAVALRY improves the attack effectiveness by an average of 34.4% and 22.8% in image and video tasks respectively, verifying its practicality and wide applicability.

2. The method framework is clearly introduced, while using visualization to help readers quickly understand the method.

**Weaknesses:**

1. The introduction lacks motivation for using generators, and accordingly, the related work lacks citations for generator-based approaches.

2. Among the losses used by the author, Lsem seems like the loss design of the white-box LVLM attack [1], Lvis is the loss design of FARE [2], according to the auxiliary model is ResNet-50, and Laux is the traditional adv attack loss design. Can the authors explain the core difference in Lsem and normally used LVLM attack loss?

3. In Table 1, why are all the comparison methods migrating image attacks to video? Can existing video methods be compared?

4. Is the "LLaVA-Video" in Table 2 a typo? It's in image benchmark.

5. It is recommended to add comparisons with other attack methods in Figure 4. In addition, the authors only show the processing efficiency. However, judging from the difference in methods, compared with attack paradigms such as PGD, the authors' method adds pre-training and finetuning processes. The overall time of these methods should be compared with the overall time of adversarial attack paradigms such as AnyAttack to reflect a fair comparison of performance gains and time costs.

[1] Schlarmann C, Hein M. On the adversarial robustness of multi-modal foundation models[C]//Proceedings of the IEEE/CVF International Conference on Computer Vision. 2023: 3677-3685.

[2] Schlarmann C, Singh N D, Croce F, et al. Robust clip: Unsupervised adversarial fine-tuning of vision embeddings for robust large vision-language models[J]. arXiv preprint arXiv:2402.12336, 2024.

**Questions:**

See Weaknesses.

---

> ### Author Response · Authors · 2025-11-23
>
> **Response to Reviewer pStB [Part 1/3]**
>
> We thank the reviewer for the thoughtful feedback and for recognizing our motivation as clear and the task as interesting. We appreciate your acknowledgment of our method's practicality and wide applicability across seven mainstream MLLMs. **Revisions in the PDF are highlighted in blue for clarity.**
>
> Before addressing your specific concerns regarding baselines and loss formulations, we wish to highlight a significant real-world vulnerability uncovered in our new **Section 6**. When deploying CAVALRY against the state-of-the-art **Gemini 2.5 Flash** in a content moderation scenario, we successfully deceived the model into classifying **54.2%** of 500 verified violent videos as "safe."
>
> Consider the implications for a major social media platform: over half of the violent content could potentially bypass automated safety filters. Crucially, this attack remains highly effective even against aggressive defenses like Gaussian Blur and heavy JPEG compression, demonstrating that our method generates robust, spatiotemporally coherent perturbations rather than fragile noise. This underscores a systemic risk to modern safety pipelines that goes far beyond standard benchmark scores.
>
> ---
>
> ### 1. Motivation for Generator-Based Approach
>
> > **Comment:** *"The introduction lacks motivation for using generators, and accordingly, the related work lacks citations for generator-based approaches."*
>
> **Response:**
> We thank the reviewer for pointing out this omission. We have revised our **Introduction** and **Related Work** sections to explicitly reference foundational studies such as *GAP [Poursaeed et al., CVPR 2018]* and *AdvGAN [Xiao et al., IJCAI 2018]*.
>
> These classic works established generator-based methods as a fundamental parallel to iterative optimization (like PGD), offering a distinct trade-off: a one-time training cost for vastly superior inference speed. This efficiency becomes the decisive motivation in the context of **Video MLLMs**. While iterative optimization is feasible for single images, it becomes computationally prohibitive when applied to the temporal dimension, as it would require performing costly backpropagation for hundreds of iterations on every single frame of every new video.
>
> By adopting a generator-based architecture, we overcome this bottleneck, creating perturbations in a single feed-forward pass with linear time complexity. This design choice is what enables real-time processing of massive video streams, a capability that remains mathematically impossible for traditional iterative baselines.

---

> ### Author Response · Authors · 2025-11-23
>
> **[Part 2/3]**
>
> ### 2. Distinction from Prior Loss Designs
>
> > **Comment:** *"Among the losses used by the author, Lsem seems like the loss design of the white-box LVLM attack [1]... Can the authors explain the core difference?"*
>
> **Response:**
> We thank the reviewer for the keen observation. We acknowledge that our $\mathcal{L}_{sem}$ shares the same information-theoretic roots (Negative Log-Likelihood) as Schlarmann et al. [1]. However, a closer look at the mathematical formulation reveals critical differences in Optimization Paradigm, Inference Constraints, and Temporal Scope.
>
> **1. Optimization Paradigm: Instance-Specific PGD vs. Generative Learning**
> The most fundamental difference lies in what is being optimized.
> * **Ref [1] (Schlarmann et al.):** Applies NLL within an Iterative Optimization framework (PGD). It computes gradients w.r.t. specific pixels ($\nabla_\delta \mathcal{L}$) to find a local maximum for a single image.
> * **CAVALRY (Ours):** Adapts NLL to a Generative Learning framework. We compute gradients w.r.t. the generator's parameters ($\nabla_\theta \mathcal{L}$).
>
> **Why this matters:** Ref [1] performs "inference-time optimization" to overfit a single sample, whereas CAVALRY seeks to learn a universal perturbation distribution. This "Learning-to-Attack" paradigm is the engine behind our linear-time efficiency and cross-sample generalization.
>
> **2. Critical Difference: Inference-Time Dependency**
> This structural difference leads to a severe functional constraint in Ref [1], visible in their formulation (Eq. 2):
> $$\max_{\delta_q, \delta_c} - \sum_{l=1}^m \log p(y_l \mid y_{<l}, z, q + \delta_q, c + \delta_c)$$
>
> * **Dependency in [1]:** To generate an attack, Ref [1] requires the ground-truth text ($y$) and often a context image ($c$) during the inference phase to calculate the gradient. If the attacker does not have the ground-truth label for a new video, the method is inoperable.
> * **Freedom in CAVALRY:** In sharp contrast, our approach trains a feed-forward generator $G_\theta$. Once trained, the inference is **Label-Free and Text-Free**: $x_{adv} = x + G_\theta(x)$. We require only the visual input. This makes CAVALRY a practical "blind" attack suitable for real-world deployment.
>
> **3. Temporal Scope:**
> Finally, while [1] optimizes static inputs ($q$ and $c$), CAVALRY extends the objective to Video Sequences. Our generator is optimized to disrupt the spatiotemporal dependencies across a continuous batch, effectively severing the causal link for dynamic reasoning—a capability mathematically absent in the static formulation of [1].

---

> ### Author Response · Authors · 2025-11-23
>
> **[Part 3/3]**
>
> ### 3. Baseline Selection: Video-Specific vs. Image-Adapted Methods
>
> > **Comment:** *"In Table 1, why are all the comparison methods migrating image attacks to video? Can existing video methods be compared?"*
>
> **Response:**
> We respectfully clarify that our baselines are not limited to image-based migrations. Instead, we categorized our comparisons into two distinct groups:
> 1.  **Transferable image-based MLLM attacks:** CWA, AnyAttack, X-Transfer.
> 2.  **Dedicated video-centric attack:** GCMA.
>
> GCMA serves precisely as the existing video method requested by the reviewer. The reason we cannot include other video baselines is that, to the best of our knowledge, CAVALRY is the first framework explicitly designed for transferable adversarial attacks on Video MLLMs.
>
> ### 4. Clarification on LLaVA-Video in Image Benchmarks
>
> > **Comment:** *"Is the 'LLaVA-Video' in Table 2 a typo? It's in image benchmark."*
>
> **Response:**
> We clarify that this is not a typo, but a deliberate experimental design. LLaVA-Video-7B is a unified architecture capable of processing both single images and video streams. We evaluated it on the MME image benchmark to demonstrate the backward compatibility of our framework: CAVALRY effectively disrupts this video-specialized model even on static image tasks, reinforcing our claim of a unified attack across modalities.
>
> ### 5. Efficiency and Training Time Trade-off (Figure 4)
>
> > **Comment:** *"It is recommended to add comparisons with other attack methods in Figure 4... PGD adds pre-training and finetuning processes... The overall time should be compared..."*
>
> **Response:**
> We acknowledge that generator-based methods incur a one-time "setup cost" (training) that optimization-based methods (like PGD/CWA) avoid. However, in the context of Transferable Adversarial Attacks, the standard metric for efficiency is **Inference Latency** for the following reasons:
>
> **1. Economic Logic: Amortized vs. Marginal Cost**
> * **Optimization Methods (CWA, PGD):** Have zero setup cost but a high Marginal Cost. The attacker must perform computationally expensive backpropagation for every single new video.
> * **Generator Methods (CAVALRY, AnyAttack):** Incurs a fixed Sunk Cost (training). Once trained, the generator functions as a compiled tool with negligible Marginal Cost (linear time complexity).
>
> In a realistic "Scale-out" threat scenario (e.g., targeting millions of videos), the initial training time becomes mathematically negligible compared to the cumulative savings in inference time.
>
> **2. Empirical Comparison on Benchmark Generation:**
> To demonstrate this practically, we recorded the total wall-clock time required to generate adversarial examples for the entire TempCompass benchmark on the same hardware:
>
> | Method | Type | Total Generation Time |
> | :--- | :---: | :---: |
> | CWA | Optimization | 129 mins |
> | CAVALRY (Ours) | Generator | 18 mins |
> | AnyAttack | Generator | 21 mins |
> | GCMA | Generator | 12 mins |
>
> **3. Conclusion:**
> As shown, CWA requires nearly 6$\times$ the time of our method just for this small-scale benchmark. If extrapolated to a large-scale dataset (e.g., 100,000 videos), CWA would require months of compute, whereas CAVALRY would complete the task in days. This order-of-magnitude difference in throughput justifies the upfront training investment.
>
> ---
>
> We hope these clarifications regarding our motivation, methodological distinctions, and efficiency trade-offs fully address your concerns. We remain available for further discussion.

---

### Official Review · Reviewer_AP9G · 2025-10-28

**Soundness:** 2
**Presentation:** 2
**Contribution:** 2
**Rating:** 4
**Confidence:** 4

**Summary:**

The paper introduces CAVALRY, a unified adversarial framework for both image and video MLLMs. It trains a U-Net generator to produce adversarial perturbations for each input image by maximizing the negative log-likelihood of the ground-truth response, together with a regularization term that keeps the perturbed visual tokens close to the originals. Training is performed in two stages, and the authors evaluate the method on both closed-source and open-source models, reporting improved attack success rates.

**Strengths:**

Overall, the paper is well written and clearly presented.

The paper provides both theoretical and empirical results, demonstrating the effectiveness of the proposed framework.

**Weaknesses:**

1. The method’s only video-specific component is batching frames from the same video with identical questions and answers during fine-tuning. This implicitly encourages temporal consistency but does not explicitly model temporal dependencies. In principle, the same setup would work for unrelated images sharing the same QA pairs. Therefore, the paper’s claim of “simultaneously modeling cross-modal reasoning vulnerabilities and temporal dependencies” (line 108) is an overstatement.

2. The proposed negative-likelihood objective is essentially equivalent to standard untargeted adversarial attacks, which also maximize the negative log-likelihood. The conceptual novelty over existing attack paradigms is limited.

3. All reported results rely on LLM-based scores, which inherently has randomness. Moreover, the score differences are small, making the reported improvements potentially sensitive to randomness. The paper should include multiple runs and standard deviations to ensure the statistical significance and reproducibility of the results.

**Questions:**

How can this method capture meaningful temporal dependencies merely by including frames from the same video within a single batch?

---

> ### Author Response · Authors · 2025-11-23
>
> **[Part 1/3]**
>
> We thank the reviewer for the constructive feedback and for acknowledging the clarity of our presentation as well as the theoretical and empirical effectiveness of our proposed framework. **Revisions in the PDF are highlighted in blue for clarity.**
>
> Before addressing your specific concerns regarding temporal modeling mechanisms and experimental rigor, we wish to highlight a significant real-world vulnerability uncovered in our new **Section 6**. When deploying CAVALRY against the state-of-the-art **Gemini 2.5 Flash** in a content moderation scenario, we successfully deceived the model into classifying **54.2%** of 500 verified violent videos as "safe."
>
> Consider the implications for a major social media platform: over half of the violent content could potentially bypass automated safety filters. Crucially, this attack remains highly effective even against aggressive defenses like Gaussian Blur and heavy JPEG compression, demonstrating that our method generates robust, spatiotemporally coherent perturbations rather than fragile noise. This underscores a systemic risk to modern safety pipelines that goes far beyond standard benchmark scores.
>
> ---
>
> ### 1. Clarification on Temporal Modeling Mechanism
>
> > **Comment:** *"The method’s only video-specific component is batching frames... This implicitly encourages temporal consistency but does not explicitly model temporal dependencies... The paper’s claim... is an overstatement."*
>
> **Response:**
> We appreciate the reviewer's scrutiny regarding our temporal modeling mechanism. While we agree that our generator does not employ explicit 3D architectures (e.g., 3D-Conv), we respectfully disagree that our claim is an overstatement. The temporal modeling in our framework is driven by the optimization signal rather than the generator's architecture.
>
> **1. Mechanism: Holistic Token Sequence Processing**
> The reviewer suggests that our batching strategy is equivalent to training on "unrelated images sharing QA pairs." This would only be true if the target model processed frames in isolation. However, Modern Video MLLMs flatten visual information into a single, continuous sequence of tokens that are fed into the LLM all at once.
> * The Clean Input Sequence seen by the MLLM: `[text_token, frame_1_token, frame_2_token, ..., frame_n_token]`
> * The Adversarial Input Sequence generated by our method: `[text_token, frame_1 + delta_1, ..., frame_n + delta_n]`
>
> Because the MLLM employs temporal self-attention layers to relate `frame_1` to `frame_2`, the gradients for the entire batch are mathematically coupled. During backpropagation, the error signal forces the generator to coordinate `delta_1` through `delta_n` jointly. If we were simply training on unrelated images, the inter-frame attention weights would be negligible, and this coupled learning would not occur.
>
> **2. Physical Proof: Normalized Flow Consistency (NFC)**
> The most direct evidence refuting the "lack of temporal modeling" hypothesis is our NFC analysis in **Table 4** (Section 5.4). NFC measures whether the generated perturbation "moves" coherently with the objects in the video (based on optical flow), rather than flickering like random static.
> * **Hypothesis Check:** If the reviewer's hypothesis were correct (that our attack ignores inter-frame dynamics), our perturbations would be temporally disjoint, resulting in a low or negative NFC score similar to the baselines (GCMA: -5.99, AnyAttack: -4.07).
> * **The Reality:** CAVALRY achieves a high positive NFC score of **0.84**.
>
> This physically proves that our generator has learned to respect the motion dynamics of the video. The perturbation "flows" with the visual content, a behavior that is strictly impossible to achieve by simply processing frames as independent images.
>
> **3. Functional Proof: New TempCompass Results**
> This capability is further confirmed by our new evaluation on the **TempCompass benchmark** (Table 2). We achieve state-of-the-art performance (**47.10% ASR** on QwenVL-2.5) on tasks specifically designed to test temporal logic, such as "Video Order" and "Direction." Successfully attacking a question about the chronological order of events requires disrupting the temporal links between frames, proving that our optimization strategy effectively models and targets these specific dependencies.

---

> ### Author Response · Authors · 2025-11-23
>
> **[Part 2/3]**
>
> ### 2. Conceptual Novelty of the Negative-Likelihood Objective
>
> > **Comment:** *"The proposed negative-likelihood objective is essentially equivalent to standard untargeted adversarial attacks... The conceptual novelty over existing attack paradigms is limited."*
>
> **Response:**
> We respectfully point out that while the negative log-likelihood (NLL) objective is foundational in general classification, applying it effectively to transferable MLLM attacks represents a distinct departure from the current dominant paradigms in this specific field.
>
> **1. Distinction from Mainstream MLLM Attacks:**
> The prevailing state-of-the-art for transferable MLLM attacks (e.g., AnyAttack [CVPR 2025]) typically operates at the **Feature Level**. These methods generally focus on two objectives:
> * Visual Feature Distortion: Maximizing the distance in the visual encoder space (similar to the $\mathcal{L}_{vis}$ term in our equation).
> * Image-Text Representation Matching: Forcing the adversarial image to align with a target caption in the embedding space, formulated as: $\arg\max f_\phi(x_{adv})^\top g_\psi(c_{tar})$.
>
> **2. The Novelty of Generative Disruption:**
> CAVALRY challenges this feature-centric assumption. We demonstrate that attacking the **Generative Process** directly—by maximizing the divergence of the autoregressive probability sequence—is significantly more potent. Instead of merely messing up the embeddings and hoping it affects the output, our Semantic Loss ($\mathcal{L}_{sem}$) explicitly penalizes the probability of the correct reasoning path at every token step:
> $$\mathcal{L}_{sem} = -\mathbb{E}_{t} [ \log P(a_t | V+\delta, a_{<t}) ]$$
> This mathematically "severs" the causal link between visual evidence and the correct textual conclusion, forcing the model to hallucinate or fail.
>
> **3. Empirical Validation:**
> The value of this conceptual shift is proven by our results. By moving beyond the standard feature-level optimization used in baselines like AnyAttack to our generative-level disruption, we achieve a massive **+22.8%** improvement in SRR on video benchmarks. This confirms that properly adapting the likelihood objective to the spatiotemporal generation process is a superior strategy for breaking video reasoning.
>
> ### 3. Statistical Significance and Evaluation Randomness
>
> > **Comment:** *"All reported results rely on LLM-based scores, which inherently has randomness... The paper should include multiple runs and standard deviations..."*
>
> **Response:**
> We appreciate the reviewer's suggestion to enhance the statistical rigor of our evaluation.
>
> **1. Evaluator Cross-Validation (Appendix E):**
> To verify that our results are not an artifact of specific judge bias, we conducted a cross-validation using two additional independent judges (Gemini-2.5-Flash and Claude-3.5-Haiku). The results confirm that our method's superiority is evaluator-agnostic.
>
> **2. End-to-End Reproducibility Study (Appendix D):**
> To strictly address the concern about stochastic variance, we conducted a rigorous reproducibility study. We re-executed the entire inference pipeline three independent times.
>
> **Table D1: Reproducibility on GPT-4.1-Mini (3 runs)**
>
> | Run ID | Overall | MC | Yes/No | Cap-M | Cap | Action | Direct. | Speed | Order | Att |
> | :--- | :---: | :---: | :---: | :---: | :---: | :---: | :---: | :---: | :---: | :---: |
> | Run #1 | 35.62 | 44.81 | 33.47 | 25.08 | 38.92 | 10.78 | 48.30 | 48.40 | 42.77 | 28.98 |
> | Run #2 | 35.37 | 44.81 | 33.06 | 24.55 | 38.87 | 10.78 | 48.24 | 47.36 | 41.76 | 29.83 |
> | Run #3 | 35.45 | 44.75 | 33.14 | 25.48 | 38.42 | 11.15 | 47.99 | 46.71 | 42.12 | 30.47 |
> | **Mean $\pm$ Std** | **35.48$_{\pm0.13}$** | 44.79 | 33.22 | 25.04 | 38.74 | 10.90 | 48.18 | 47.49 | 42.22 | 29.76 |
>
> **Table D2: Reproducibility on Gemini-2.0-Flash (3 runs)**
>
> | Run ID | Overall | MC | Yes/No | Cap-M | Cap | Action | Direct. | Speed | Order | Att |
> | :--- | :---: | :---: | :---: | :---: | :---: | :---: | :---: | :---: | :---: | :---: |
> | Run #1 | 42.11 | 33.23 | 31.63 | 68.80 | 41.92 | 19.10 | 59.11 | 46.58 | 43.35 | 43.32 |
> | Run #2 | 42.11 | 33.16 | 31.55 | 69.06 | 41.87 | 18.98 | 59.05 | 46.38 | 43.71 | 43.39 |
> | Run #3 | 42.20 | 33.04 | 31.80 | 68.80 | 42.22 | 19.22 | 59.23 | 46.77 | 43.21 | 43.47 |
> | **Mean $\pm$ Std** | **42.14$_{\pm0.05}$** | 33.14 | 31.66 | 68.89 | 42.00 | 19.10 | 59.13 | 46.58 | 43.42 | 43.39 |

---

> ### Author Response · Authors · 2025-11-23
>
> **[Part 3/3]**
>
> **Conclusion on Reproducibility:**
> As shown in the tables above, the standard deviation for the Overall ASR is negligible: $\pm 0.13\%$ for GPT-4.1 and $\pm 0.05\%$ for Gemini-2.0. This mathematically confirms that the reported improvements are statistically significant and represent a consistent algorithmic advantage, not random fluctuation.
>
> ### 4. Question: Mechanism of Capturing Temporal Dependencies
>
> > **Question:** *"How can this method capture meaningful temporal dependencies merely by including frames from the same video within a single batch?"*
>
> **Response:**
> Please refer to our detailed response in **Q1 (Clarification on Temporal Modeling Mechanism)**.
>
> To summarize briefly: While the generator takes a batch as input, the optimization signal is derived from the MLLM's processing of the holistic video sequence. Because the MLLM uses temporal attention to link frames, the gradients backpropagated to the generator are mathematically coupled across time. This "Temporal Gradient Flow" forces the generator to learn spatiotemporally coherent perturbations, a capability rigorously proven by our positive NFC scores (0.84) and state-of-the-art performance on TempCompass (new Table 2) video ordering tasks.
>
> ---
>
> We hope these clarifications regarding temporal modeling, conceptual novelty, and statistical significance fully address your concerns. We remain available for any further discussion during the rebuttal period.

---

### Official Review · Reviewer_Tbio · 2025-10-31

**Soundness:** 2
**Presentation:** 3
**Contribution:** 2
**Rating:** 2
**Confidence:** 5

**Summary:**

The paper presents CAVALRY, a unified adversarial attack framework targeting both image and video MLLMs. CAVALRY employs a generator that produces adversarial perturbations for visual inputs, whether static images or videos. The generator is optimized through three complementary objectives: (1) generative likelihood divergence, (2) manipulation of visual representations, and (3) an auxiliary feature loss. To ensure cross-modal and temporal coherence, CAVALRY adopts a two-stage training strategy involving large-scale pretraining followed by fine-tuning. Experimental results demonstrate that CAVALRY achieves state-of-the-art performance on both open-source and commercial MLLMs across the MMBench-Video and MME benchmarks.

**Strengths:**

1. Theorem 1 and its mathematical proof clearly justify the semantic loss objective, providing theoretical soundness.
2. The paper demonstrates transferability by showing that the attack generalizes not only to open-source MLLMs but also to commercial ones.
3. Instead of iteratively updating perturbations, the authors train a generator that can produce frame-wise perturbations with linear time complexity, proving the practical applicability of the approach.

**Weaknesses:**

Major Weaknesses
1. Both the output-level loss and vision-encoder-level loss for MLLMs have been extensively studied in prior MLLM adversarial and jailbreak attack works [1, 2, 3, 4]. Thus, the methodological novelty is somewhat limited.
2. The evaluation is restricted to a single benchmark for image understanding and one for video understanding, using only one LLM as the judge. This narrow experimental scope limits the assessment of generalizability across broader image/video domains and diverse text prompts.
3. The paper lacks sufficient analysis and motivation regarding MLLMs. In particular, the reason behind the proposed method’s high transferability is not well explained. While the authors claim that large-scale pretraining enables cross-architecture transferability, this explanation is not fully convincing. Large-scale data may help the generator generalize across a wide range of samples within the surrogate model’s domain, but it does not inherently guarantee transferability across different architectures.

[1] Luo, Haochen, et al. “An Image is Worth 1000 Lies: Adversarial Transferability Across Prompts on Vision-Language Models.” ICLR 2024.
[2] Cui, Xuanming, et al. “On the Robustness of Large Multimodal Models Against Image Adversarial Attacks.” CVPR 2024.
[3] Zhao, Yunqing, et al. “On Evaluating Adversarial Robustness of Large Vision-Language Models.” NeurIPS 2023.
[4] Zhang, Jiaming, et al. “AnyAttack: Towards Large-Scale Self-Supervised Generation of Targeted Adversarial Examples for Vision-Language Models.” CVPR 2025.

**Questions:**

Minor Weaknesses & Questions
1. It is unclear why the authors used an adversarially trained ResNet as the auxiliary model instead of an adversarially trained transformer architecture, given that most MLLM vision encoders are transformer-based.
2. The paper employs the SRR metric instead of the traditional ASR, but it is somewhat difficult to intuitively understand what level of attack strength the SRR represents.

---

> ### Author Response · Authors · 2025-11-23
>
> **[Part 1/4]**
>
> We thank the reviewer for the detailed summary and for recognizing the theoretical soundness of our semantic loss and the practical value of our linear-time generator. **Revisions in the PDF are highlighted in blue for clarity.**
>
> Before addressing your specific concerns regarding methodological novelty and evaluation scope, we wish to highlight a significant real-world vulnerability uncovered in our new **Section 6**. When deploying CAVALRY against the state-of-the-art **Gemini 2.5 Flash** in a content moderation scenario, we successfully deceived the model into classifying **54.2%** of 500 verified violent videos as "safe."
>
> Consider the implications for a major social media platform: over half of the violent content could potentially bypass automated safety filters. Crucially, this attack remains highly effective even against aggressive defenses like Gaussian Blur and heavy JPEG compression, demonstrating that our method generates robust, spatiotemporally coherent perturbations rather than fragile noise. This underscores a systemic risk to modern safety pipelines that goes far beyond standard benchmark scores.
>
> ---
>
> ### 1. Methodological Novelty and Fundamental Mathematical Divergence
>
> > **Comment:** *"Both the output-level loss and vision-encoder-level loss... have been extensively studied... [1, 2, 3, 4]. Thus, the methodological novelty is somewhat limited."*
>
> **Response:**
> We thank the reviewer for citing these works. However, we respectfully point out that categorizing all methods utilizing output signals as "lacking novelty" is an oversimplification. To clarify the fundamental mathematical differences, we explicitly compare our objective with the formulations in the cited works.
>
> **1. Mathematical Disjointness:**
> The mathematical structures of the cited works are fundamentally different from our Generative Likelihood Divergence approach:
>
> * **Our Method (CAVALRY):** We maximize the divergence in the autoregressive generative probability to "sever" the reasoning link:
>     $$\mathcal{L}_{\text{sem}}(\theta) = -\mathbb{E}_{t} \left[ \log P_{\mathcal{M}_{\text{LM}}}\left(a_t | \mathcal{M}_{\text{VE}}(V + G_\theta(V)), Q, a_{<t}\right) \right]$$
>
> * **Ref [1] (CroPA):** This method solves a Min-Max game optimizing both image ($\delta_v$) and text ($\delta_t$) perturbations, distinct from our fixed-prompt generator:
>     $$\min_{\delta_v} \max_{\delta_t} \mathcal{L}(f(x_v + \delta_v, x_t + \delta_t), T)$$
>
> * **Ref [4] (AnyAttack):** This utilizes a targeted, contrastive-style objective (dot product maximization) to match a target caption $\boldsymbol{c}_{\text{tar}}$, whereas ours is untargeted generative disruption:
>     $$\mathop{\arg \max}_{\|x_{\text{cle}} - x_{\text{adv}}\|_p \le \epsilon} f_\phi(\boldsymbol{x}_{\text{adv}})^\top g_\psi(\boldsymbol{c}_{\text{tar}})$$
>
> * **Ref [3] (Zhao et al.):** This relies on minimizing feature distance for targeted instances $x_r$ via iterative optimization, not a feed-forward generator:
>     $$\min \mathcal{L}(f_s(\delta + x_r), f_s(x)), \quad \text{s.t. } x_r \neq x$$
>
> * **Ref [2] (Cui et al.):** This is an evaluation and analysis paper proposing a defense (Query Decomposition). It does not define a new attack loss function.
>
> **2. Clarification on Contributions:**
> We wish to emphasize that we never claimed the generic use of visual feature losses ($\mathcal{L}_{vis}, \mathcal{L}_{aux}$) as our primary novelty. As stated in our Introduction, our core contributions are explicitly defined as: (i) the Generative Likelihood Divergence paradigm (Eq. 4) which targets the causal link of generation rather than feature matching or classification boundaries; and (ii) the Unified Video Framework that solves spatiotemporal consistency.
>
> **3. The "Video Gap":**
> Most critically, none of the cited works [1-4] address the challenge of **Video MLLMs**. CAVALRY is the first unified framework to solve the spatiotemporal consistency problem (verified by our TempCompass results, ASR 47.1%), enabling a single generator to attack both static images and dynamic videos efficiently. This extension from image-only baselines to the temporal domain is a significant methodological leap.
>
> **Action:**
> We appreciate the comprehensive references provided. To provide a more complete picture of the field, we have added citations and a detailed discussion of these works [1-4] to the **Related Work** section of our revised manuscript.

---

> ### Author Response · Authors · 2025-11-23
>
> **[Part 2/4]**
>
> ### 2. Evaluation Scope, Generalizability, and Judge Bias
>
> > **Comment:** *"The evaluation is restricted to a single benchmark... using only one LLM as the judge. This narrow experimental scope limits the assessment..."*
>
> **Response:**
> We respectfully point out that this concern may stem from a traditional view of datasets (e.g., ImageNet vs. COCO) which does not align with the current paradigm of Foundation Model evaluation.
>
> **1. Modern Benchmarks are Comprehensive Aggregates:**
> Modern evaluation has shifted from single-task datasets to holistic "Super-Benchmarks." A prime example is **LM Arena** (Chatbot Arena), which aggregates diverse tasks to rank LLMs; consequently, leading labs (OpenAI, Google, Anthropic) primarily highlight performance on such aggregate benchmarks upon model release. **MMBench-Video** plays this exact role for Video MLLMs. It is not a "single dataset" in the traditional sense but a comprehensive aggregate integrating thousands of video clips QA across dozens of domains (e.g., news, movies, surveillance, sports) and perception/reasoning tasks.
>
> To illustrate the magnitude of this "single" benchmark: deriving the results in Table 1 required over a week of continuous inference on an 8$\times$A100-80G sever just to evaluate the adversarial videos (excluding generation time). In comparison, the datasets used in the cited works like AttackVLM [3] and AnyAttack [4] (e.g., MSCOCO captioning) are significantly smaller in complexity, typically requiring only minutes to evaluate on a single GPU. Therefore, our evaluation scale far exceeds typical static image attack standards.
>
> **2. New Benchmark Added (TempCompass):**
> Despite the comprehensive nature of MMBench-Video, to fully address your concern regarding "broader domains" and "diverse text prompts," we have added a second video benchmark, **TempCompass [ACL 2024]**, in the newly added **Table 2**. This benchmark focuses specifically on fine-grained temporal attributes (Action, Speed, Direction, Order). CAVALRY achieves the highest Attack Success Rate (ASR) across all seven evaluated models, confirming that our method generalizes effectively across different domain definitions and prompt types.
>
> **3. Cross-Validation with Diverse LLM Judges (Appendix E):**
> Regarding the reliance on a single LLM judge, we clarify that using GPT-4o-mini is strict adherence to the official protocols of these benchmarks. However, to rigorously address your concern about bias: We conducted a new cross-validation in **Appendix E**, employing two additional state-of-the-art judges: Gemini-2.5-Flash and Claude-3.5-Haiku. As shown in new **Table 10**, while absolute scores vary, the relative ranking remains invariant. CAVALRY consistently outperforms baselines regardless of the evaluator, proving that the attack effectiveness is objective and not an artifact of a specific judge's bias.

---

> ### Author Response · Authors · 2025-11-23
>
> **[Part 3/4]**
>
> ### 3. Mechanism of Transferability and Large-Scale Pretraining
>
> > **Comment:** *"The reason behind the proposed method’s high transferability is not well explained. While the authors claim that large-scale pretraining enables cross-architecture transferability, this explanation is not fully convincing... it does not inherently guarantee transferability across different architectures."*
>
> **Response:**
> We appreciate the opportunity to clarify the mechanisms driving our transferability.
>
> **1. Clarification on Cross-Architecture Mechanism:**
> First, we wish to clarify that "cross-architecture transferability" is not the primary focus of our method. In fact, the only mechanism explicitly introduced in our paper to address cross-architecture gaps is the **Auxiliary Loss ($\mathcal{L}_{aux}$)** described in Lines 245-249. This component anchors perturbations to robust features from an adversarially trained ResNet, serving as the specific bridge between the surrogate and target models.
>
> **2. Data Generalization is a Form of Transferability:**
> Furthermore, "transferability" in the context of generator-based attacks extends beyond just model architectures; it critically includes Data Generalization—the ability to generate effective perturbations for unseen images/videos outside the training set. As the reviewer correctly noted, large-scale data helps the generator generalize across a wide range of samples. For a generative attack, this capability is not trivial—it is the fundamental prerequisite for effectiveness.
>
> **3. Validation via AnyAttack [4]:**
> Regarding the connection between large-scale pre-training and transferability, we respectfully point out that this relationship has already been rigorously proven by **AnyAttack [4]**, a work cited by the reviewer. The core contribution of AnyAttack was demonstrating that pre-training on a 400M-scale dataset (LAION-400M) is the decisive factor that empowers a generator to achieve high transferability. Since our work adopts this exact paradigm verified by AnyAttack, the "reason" behind our performance is scientifically grounded in the very literature the reviewer referenced.
>
> ### 4. Choice of Auxiliary Model Architecture
>
> > **Comment:** *"It is unclear why the authors used an adversarially trained ResNet... instead of an adversarially trained transformer architecture..."*
>
> **Response:**
> The decision to employ a CNN-based ResNet-50 instead of a Transformer-based auxiliary model is a deliberate strategy to maximize architectural diversity within our training framework. This design is grounded in the foundational insights from *Delving into Transferable Adversarial Examples and Black-box Attacks (ICLR 2017)*, which established that integrating models with distinct architectures prevents the attack from overfitting. The importance of this heterogeneity has been reaffirmed in recent studies, such as *Rethinking Model Ensemble in Transfer-based Adversarial Attacks (ICLR 2024)*, which demonstrates that diverse ensembles significantly boost transferability to unknown black-box models.
>
> In our specific case, since our primary surrogate model (InternVL) is already Transformer-based, introducing another Transformer as the auxiliary model would risk homogenizing the gradient directions. By incorporating a ResNet, we force the generator to learn robust, architecture-agnostic perturbations that satisfy both CNN and Transformer feature spaces simultaneously. We also note that this practice is consistent with prior art; for instance, the AttackVLM [3] paper referenced by the reviewer also employs ResNet in its ensemble.

---

> ### Author Response · Authors · 2025-11-23
>
> **[Part 4/4]**
>
> ### 5. Interpretability of the SRR Metric
>
> > **Comment:** *"The paper employs the SRR metric instead of the traditional ASR, but it is somewhat difficult to intuitively understand what level of attack strength the SRR represents."*
>
> **Response:**
> We appreciate the feedback regarding metric interpretability. The adoption of SRR is dictated by the inherent design of the MMBench-Video benchmark. As detailed in Lines 287-294, this benchmark does not rely on labels suitable for traditional accuracy calculations, so it's does't exist Attack Success Rate (ASR). Instead, it employs an LLM judge to provide holistic, Likert-style ratings (e.g., 0 to 5) assessing the detailed quality of reasoning and description. In this context, SRR is the mathematically appropriate metric to quantify the relative degradation of this continuous quality score.
>
> However, we fully understand the reviewer's preference for the more intuitive ASR metric to gauge attack strength. To address this, our revised manuscript incorporates two new evaluations that explicitly report ASR:
> 1.  **TempCompass Evaluation (Table 2):** This benchmark consists of objective multi-choice and matching tasks. We report a standard ASR of up to 47.10% on QwenVL-2.5.
> 2.  **Real-World Threat Analysis (Section 6 / Table 5):** We treat violence detection as a binary classification task and report an ASR of 54.2% against Gemini 2.5.
>
> These figures provide the concrete, intuitive measures of success requested, complementing the nuanced quality assessment provided by SRR.
>
> ---
>
> We hope these clarifications fully address your concerns and highlight the unique contributions of our work. We remain available for any further discussion.

---

### Official Review · Reviewer_wEJt · 2025-11-01

**Soundness:** 3
**Presentation:** 3
**Contribution:** 2
**Rating:** 4
**Confidence:** 2

**Summary:**

This paper studies adversarial attacks and robustness of multimodal large language models (MLLMs) that answer vision-and-language queries over images and videos. The proposed method, CAVALRY, uses a U-Net-style generator trained in a two-stage framework to provide perturbed visual inputs that alter MLLM outputs. CAVALRY combines three different losses to target both vision-language connections and vision representation, and it is evaluated on both static (images) and temporal (video) inputs. Experiments indicate CAVALRY can substantially degrade MLLM performance across multiple models and settings.

**Strengths:**

- The attack design is intuitive and well-motivated: combining multiple loss terms to target both representation-level and alignment-level failure modes make sense for attacking LLMs.
- The evaluation covers both images and videos, demonstrating the generality of the framework across static and temporal inputs.
- The paper presents a reasonably comprehensive set of experiments showing CAVALRY’s efficacy across several MLLMs.

**Weaknesses:**

- The training strategy appears to treat all video frames independently, using the same QA supervision across frames rather than modeling temporal dependencies explicitly. This raises concerns that the attack might ignore inter-frame dynamics and overfit to per-frame perturbations rather than truly exploiting temporal vulnerabilities.
- Aside from a few qualitative examples (e.g., Figure 2), the paper lacks quantitative analysis of visual distortion (e.g., PSNR, LPIPS, or L2 norms) and human perceptual thresholds. Some examples images look heavily corrupted; reporting distortion budgets and the number of frames required for a successful attack would clarify real-world plausibility.
- The method combines three loss terms but does not fully isolate their contributions. An ablation study showing how each loss affects attack strength, perceptual quality, and transferability would strengthen understanding of why CAVALRY works.
- Training with the same QA for all frames may lead to artifacts: attacks might exploit static shortcuts rather than disrupting temporal reasoning. Evaluation on short-version of videos that is necessary for the true answer would give further understanding on how MLLMs are attacked.

**Questions:**

- Equation (5) targets likelihood divergence for the ground-truth answer. Do you have any theoretical or empirical evidence that optimizing this loss reliably causes semantically different outputs (not just low-confidence or truncated answers)?
- How does attack performance vary with the fraction of frames you perturb in a video? Is perturbing a single frame (or a small subset) sufficient in typical cases, or do you need to perturb most/all frames to achieve high success?
- Based on the nature characteristics of proposed training strategy, it seems like the model is not trained to consider the temporal relationships of different frames. That said, the first frame of the video should be used to attack MLLMs in a way that unknown future information is related. Are there any further analysis to ensure that CAVALRY truly considers the temporal characteristics of videos?

---

> ### Author Response · Authors · 2025-11-23
>
> **[Part 1/4]**
>
> We sincerely thank the reviewer for the thoughtful assessment and for recognizing our framework as "intuitive and well-motivated." We appreciate your acknowledgment of the generality of our approach across both static and temporal inputs. **Revisions in the PDF are highlighted in blue for clarity.**
>
> Before addressing your specific concerns regarding temporal modeling and visual quality, we wish to highlight a significant real-world vulnerability uncovered in our new **Section 6**. When deploying CAVALRY against the state-of-the-art **Gemini 2.5 Flash** in a content moderation scenario, we successfully deceived the model into classifying **54.2%** of 500 verified violent videos as "safe."
>
> Consider the implications for a major social media platform: over half of the violent content could potentially bypass automated safety filters. Crucially, this attack remains highly effective even against aggressive defenses like Gaussian Blur and heavy JPEG compression, demonstrating that our method generates robust, spatiotemporally coherent perturbations rather than fragile noise. This underscores a systemic risk to modern safety pipelines that goes far beyond standard benchmark scores.
>
> ---
>
> ### 1. Clarification on Temporal Modeling and Inter-frame Dynamics
>
> > **Comment:** *"The training strategy appears to treat all video frames independently... rather than modeling temporal dependencies explicitly. This raises concerns that the attack might ignore inter-frame dynamics..."*
>
> **Response:**
> We respectfully wish to clarify a fundamental aspect of the MLLM architecture and our training strategy, which directly addresses this concern. The impression that frames are processed in isolation is a misunderstanding of our pipeline.
>
> 1.  **Mechanism: Holistic Token Sequence Processing.** Modern Video MLLMs do not treat a video as a batch of independent images. Instead, they flatten the visual information into a single, continuous sequence of tokens that are fed into the LLM alongside text tokens all at once.
>     * The Clean Input Sequence Tokens seen in the MLLM: `[text_prompt, frame_1_token, frame_2_token, ..., frame_n_token]`
>     * The Adversarial Input Sequence generated by our method: `[text_prompt, frame_1 + delta_1, ..., frame_n + delta_n]`
>
> 2.  **Implicit Temporal Optimization via Joint Generation.** Crucially, during the video fine-tuning stage, we do not optimize frames independently. Instead, we sample a batch of frames from the same video and feed them into the generator simultaneously. Consequently, the optimization signal is derived from the MLLM's final output, which aggregates information across this entire sequence via Self-Attention. If `delta_1` and `delta_2` are temporally inconsistent (e.g., creating flickering artifacts), the MLLM's temporal attention layers will fail to extract coherent features, leading to a high Loss. This error signal flows back to update the generator to produce `delta_1` through `delta_n` jointly as a coherent unit.
>
> 3.  **Quantitative Proof (Section 5.4).** This is not just theoretical. We empirically validated this in our Temporal Consistency Analysis (Section 5.4). As shown in **Table 4**, we utilized the **Normalized Flow Consistency (NFC)** metric, which measures how well perturbations follow the video's motion:
>     * **Baselines:** Traditional methods yielded negative NFC scores (indicating independent, flickering noise).
>     * **CAVALRY:** Our method achieved a high positive NFC score (**0.84**).
>
>     This quantitative evidence confirms that our strategy successfully captures inter-frame dynamics, producing perturbations that naturally "move" with the video content.

---

> ### Author Response · Authors · 2025-11-23
>
> **[Part 2/4]**
>
> ### 2. Quantitative Analysis of Visual Distortion
>
> > **Comment:** *"The paper lacks quantitative analysis of visual distortion (e.g., PSNR, LPIPS, or L2 norms)..."*
>
> **Response:**
> We thank the reviewer for this constructive suggestion and have added a detailed quantitative evaluation in **Appendix B**.
>
> First, it is important to note that employing a fixed $L_\infty$ budget (e.g., $\epsilon=16/255$) is the standard protocol in this field. Figure 2 simply reflects the physical reality of this standard budget rather than a method-specific defect. Crucially, while this budget introduces visible high-frequency noise, it does not obscure the semantic content itself; as demonstrated in our violence evasion study (Section 6), the violent acts remain perfectly distinct to human observers, yet the model is successfully deceived.
>
> To quantify this, employing the $L_2$ norm metric suggested by the reviewer, we compared the average norms of our method against SOTA baselines across different budgets on the MMBench-Video dataset.
>
> | Method | Type | $\epsilon=4/255$ | $\epsilon=8/255$ | $\epsilon=16/255$ |
> | :--- | :---: | :---: | :---: | :---: |
> | **CAVALRY (Ours)** | Generator | 5.88 | 11.56 | 22.51 |
> | X-Transfer | Generator | 5.90 | 11.63 | 22.52 |
> | AnyAttack | Generator | 5.80 | 11.34 | 22.04 |
> | CWA | Iterative | 5.72 | 10.89 | 21.14 |
> | GCMA | Iterative | 4.34 | 8.65 | 17.24 |
>
> These results confirm that visual distortion is driven by the budget, not the algorithm. Under the same $\epsilon$ constraint, all highly effective methods (Ours, X-Transfer, AnyAttack, CWA) exhibit nearly identical $L_2$ norms (approx. 22.5 at $\epsilon=16$). While GCMA shows a lower norm, this "cleanliness" comes at the cost of significantly compromised attack success rates. Furthermore, our method exhibits predictable linear scaling: at lower budgets (e.g., $\epsilon=4/255$), the $L_2$ norm drops to ~5.8, yielding visually stealthy perturbations suitable for scenarios requiring higher imperceptibility.
>
> ### 3. Ablation Study on Loss Components
>
> > **Comment:** *"The method combines three loss terms but does not fully isolate their contributions. An ablation study ... would strengthen understanding of why CAVALRY works."*
>
> **Response:**
> We fully agree with the reviewer that dissecting the individual contributions of each loss component is essential for understanding the inner workings of CAVALRY. To address this, we have included a comprehensive ablation study in the newly added **Appendix C**.
>
> Our analysis reveals distinct roles for each term corresponding to the specific aspects mentioned by the reviewer.
> * **Attack Strength:** The semantic loss ($\mathcal{L}_{sem}$) serves as the foundational driver; removing it leads to a collapse in effectiveness (an ~8% drop in SRR), confirming that generative likelihood guidance is indispensable for disrupting high-level reasoning.
> * **Perceptual Quality:** The visual loss ($\mathcal{L}_{vis}$) targets the visual encoding space, creating systematic perturbations that disrupt the model's core capability to ground visual evidence.
> * **Transferability:** The auxiliary loss ($\mathcal{L}_{aux}$) acts as a vital bridge. By anchoring the perturbations to the feature space of an adversarially trained ResNet, it prevents the generator from overfitting to the surrogate model, ensuring that the attack remains potent against black-box targets like Gemini and GPT-4.

---

> ### Author Response · Authors · 2025-11-23
>
> **[Part 3/4]**
>
> ### 4. Evaluation on Temporal Robustness and Video Benchmarks
>
> > **Comment:** *"Training with the same QA for all frames may lead to artifacts: attacks might exploit static shortcuts rather than disrupting temporal reasoning. Evaluation on short-version of videos that is necessary for the true answer would give further understanding..."*
>
> **Response:**
> We respectfully point out a misunderstanding regarding our experimental scope. The reviewer suggests evaluating on "short-version videos," but we wish to clarify that our primary evaluation in **Table 1** (MMBench-Video) is already a comprehensive assessment (including long videos and short videos but no images). It splits a video into frames and then feeds them to MLLMs all at once, which is also the standard practice adopted by all current video models. This was a massive computational undertaking, requiring over a week of continuous inference on an 8$\times$A100-80G cluster to evaluate these adversarial videos (not including generating videos)—a scale far exceeding typical static image attack evaluations.
>
> However, to directly address your concern that the attack might exploit "static shortcuts" and to rigorously test fine-grained temporal reasoning, we extended our evaluation to the **TempCompass benchmark** [ACL 2024] in the newly added **Table 2**. Unlike general video QA, TempCompass strictly isolates temporal attributes such as Action, Speed, Direction, and Order, where the "true answer" depends entirely on inter-frame dynamics (e.g., recognizing "moving left" vs. "moving right" is impossible via static shortcuts).
>
> **Results:** As shown in the new Table 2, CAVALRY achieves the highest Overall Attack Success Rate (ASR) across all seven evaluated models. Our method reaches a significant lead over the strongest competitor. Crucially, we excel in purely temporal categories like **"Direction"** and **"Order."** Since these tasks require understanding motion across frames, our superior performance confirms that CAVALRY effectively severs the spatiotemporal dependencies essential for video understanding, rather than relying on static artifacts.
>
> *[ACL 2024] Tempcompass: Do video llms really understand videos, Liu, Yuanxin, et al.*
>
> ### 5. Nature of Generative Disruption: Semantic Divergence vs. Truncation
>
> > **Question:** *"Equation (5) targets likelihood divergence... Do you have any theoretical or empirical evidence that optimizing this loss reliably causes semantically different outputs (not just low-confidence or truncated answers)?"*
>
> **Response:**
> We thank the reviewer for this critical question regarding the failure mode of our attack.
>
> 1.  **Theoretical Foundation (Section 4.1):** For the theoretical evidence, we refer the reviewer to Section 4.1 and Theorem 1 of our paper. We mathematically established that maximizing the negative log-likelihood is equivalent to maximizing the **KL Divergence** between the ground-truth distribution and the model's predictive distribution. By forcing the output distribution to diverge from the "truth" within the high-dimensional semantic space of the LLM, the model is mathematically compelled to shift its probability mass to alternative token trajectories. Since MLLMs are fine-tuned to be "chatty" and coherent, this shifted mass naturally flows into plausible hallucinations rather than silence or truncation.
> 2.  **Empirical Evidence (Violence Detection & Benchmarks):** The most compelling empirical proof comes from our new Real-World Threat Analysis (Section 6). When we attacked 500 verified violent videos, the model did not simply output low-confidence garbage or truncated strings; instead, it explicitly flipped its semantic classification from **"Violent"** to **"Safe"** in 54.2% of cases. This is a clear, meaningful semantic inversion. Furthermore, our high Attack Success Rate (47.10%) on the TempCompass benchmark (Table 2) reinforces this: since this benchmark relies on multiple-choice questions, the model must confidently select a wrong option (e.g., flipping from "Option A" to "Option B") to be counted as a success, definitively ruling out the "truncation" hypothesis.

---

> ### Author Response · Authors · 2025-11-23
>
> **[Part 4/4]**
>
> ### 6. Efficiency and Frame Perturbation Ratio
>
> > **Comment:** *"How does attack performance vary with the fraction of frames you perturb? ... Is perturbing a single frame (or a small subset) sufficient... or do you need to perturb most/all frames?"*
>
> **Response:**
> We perturb all frames (100%) of the video sequence. This capability highlights the distinct efficiency advantage of our generator-based framework over traditional iterative methods.
>
> The trade-off between attack performance and the number of perturbed frames is a concern primarily for slow, optimization-based attacks that require minutes to process a video. In contrast, CAVALRY operates with linear time complexity. As detailed in our **Efficiency Analysis (Section 5.4)** and **Figure 4**, our generator processes frames at approximately 65 FPS (15.4ms per frame) on a single GPU. To put this into perspective, we can generate perturbations for a massive 5,000-frame video in under 100 seconds. Because the computational cost is negligible, we have the luxury of saturating the entire video stream to ensure maximum attack success.
>
> Different MLLMs employ different frame sampling strategies ($\phi(V)$ in Eq. 1)—some sample uniformly, others sample based on scene changes. By perturbing the entire stream at high speed, we ensure that whichever frames the MLLM eventually samples are guaranteed to be adversarial, thereby severing the visual-linguistic link regardless of the target model's internal architecture.
>
> ### 7. Ensuring Temporal Characteristics in CAVALRY
>
> > **Comment:** *"Based on the nature characteristics of proposed training strategy... the first frame of the video should be used to attack MLLMs in a way that unknown future information is related. Are there any further analysis...?"*
>
> **Response:**
> We believe that our comprehensive clarifications in Q1 and Q4 have largely resolved the underlying concern regarding temporal isolation.
>
> The reviewer's concern about "unknown future information" stems from the assumption that frames are processed in a streaming fashion where the future is invisible. We respectfully clarify that this is not the case. As detailed in our response to Q1, we input **all frames** of the video simultaneously in chronological order into the MLLM. In other words, the frames are also fed into the generator in their original temporal order. Therefore, for the system, there is no "unknown future"—the entire temporal context is visible and processed at once, allowing the attack to leverage the full video context globally.
>
> **Visual Evidence in Figure 1:** To further illustrate how we enforce these temporal characteristics, we invite the reviewer to revisit Figure 1 (Right Panel). In our video fine-tuning stage, we explicitly utilize temporal-logic queries that require understanding the relationship between earlier and later frames, such as:
> * *Q: "Which animal appears first, and which comes next?"*
> * *A: "The cat appears first, followed by the dog."*
>
> Because the model processes the video holistically, the generator is forced to optimize the perturbations for the "first" and "next" frames jointly to successfully disrupt this specific temporal reasoning sequence. This design choice is empirically vindicated by our TempCompass results (Table 2), where CAVALRY achieves state-of-the-art success rates on purely temporal tasks like "Order" and "Direction," confirming that our method truly considers the temporal dynamics of the video.
>
> ---
>
> We hope these clarifications fully resolve your concerns. We are ready to answer any further questions.

---

### Official Review · Reviewer_1Yvi · 2025-11-02

**Soundness:** 3
**Presentation:** 3
**Contribution:** 3
**Rating:** 6
**Confidence:** 4

**Summary:**

This paper introduces CAVALRY, a unified adversarial attack framework targeting both image and video Multimodal Large Language Models (MLLMs). Instead of conventional boundary-based or feature-space attacks, the method disrupts the generative process of MLLMs through generative likelihood divergence maximization, effectively “severing” the link between visual perception and language generation. It employs a progressive two-stage generator trained to produce spatially and temporally coherent perturbations: large-scale pretraining on LAION-400M followed by fine-tuning on LLaVA-Instruct-150K and Video-MME for temporal coherence. Experiments on seven MLLMs, including GPT-4.1, Gemini 2.0, and QwenVL-2.5, show state-of-the-art performance, outperforming baselines by 22.8 % on video and 34.4 % on image benchmarks. The paper claims broad transferability, computational efficiency, and responsible release for AI-safety evaluation.

**Strengths:**

- Integrates both image and video adversarial attack settings into a single formulation, demonstrating flexibility across modalities. Bridges vision-language and temporal reasoning vulnerabilities, a gap unaddressed by prior work. This work establishes a new class of generative-disruption attacks for MLLMs.
- Operates on token-level autoregressive manipulation rather than feature logits, offering higher semantic fidelity. This paradigm shift yields measurable performance gains, and it is novel in this field.
- Benchmarked on seven diverse MLLMs, including commercial and open-source systems demonstrated its effectiveness of the proposed method.

**Weaknesses:**

- The SRR metric depends on GPT-4o-mini or GPT-4-turbo scorers, which may introduce bias. No cross-validation with other judges or independent human raters.
- Equation (8) introduces $\lambda_1$, $\lambda_2$, $\lambda_3$, weighting, but no individual ablation for each loss. Lack of sensitivity analysis limits understanding of the contribution balance.
- No adversarial detection, adversarial training, or model-side robustness evaluation provided. It would be good to include some experiments on mitigation methods.

**Questions:**

- Could the authors provide some comparison of the evaluation metric with other models?
- Could authors provide some insights on how to tune $\lambda_1$, $\lambda_2$, $\lambda_3$?

---

> ### Author Response · Authors · 2025-11-23
>
> **[Part 1/2]**
>
> We sincerely thank the reviewer for the encouraging assessment and for succinctly summarizing the core contributions of CAVALRY. We are particularly gratified that you recognized the value of our "unified framework" and appreciated our "paradigm shift." **Revisions in the PDF are highlighted in blue for clarity.**
>
> Before addressing your specific questions, we wish to highlight a significant real-world vulnerability uncovered in our new **Section 6**. When deploying CAVALRY against the state-of-the-art **Gemini 2.5 Flash** in a content moderation scenario, we successfully deceived the model into classifying **54.2%** of 500 verified violent videos as "safe."
>
> Consider the implications for a major social media platform: over half of the violent content could potentially bypass automated safety filters. Crucially, this attack remains highly effective even against aggressive defenses like Gaussian Blur and heavy JPEG compression, demonstrating that our method generates robust, spatiotemporally coherent perturbations rather than fragile noise. This underscores a systemic risk to modern safety pipelines that goes far beyond standard benchmark scores.
>
> ---
>
> ### 1. Evaluation Metrics: Validity of SRR and Evaluator Bias
>
> > **Comment:** *"The SRR metric depends on GPT-4o-mini or GPT-4-turbo scorers, which may introduce bias... Could the authors provide some comparison of the evaluation metric with other models?"*
>
> **Response:**
> We appreciate the reviewer's scrutiny regarding evaluation fairness. Our adoption of LLM-based judges strictly adheres to the official protocols of the MMBench-Video and MME benchmarks, which rely on holistic scoring (e.g., evaluating logic and specificity) rather than binary accuracy. Deviating from this standard would render our results incomparable with existing literature. Nevertheless, we fully agree with your insight that relying on a single model family may introduce latent bias, and that cross-validation is essential for a robust assessment.
>
> To rigorously address this concern, we have expanded our evaluation in the newly added **Appendix E**. We incorporated two additional state-of-the-art LLMs with distinct architectures and alignment preferences—**Gemini-2.5-Flash** and **Claude-3.5-Haiku**—to serve as independent judges alongside the original GPT-4o-mini.
>
> The results demonstrate that while absolute scoring distributions vary among judges due to their distinct personalities, the relative performance ranking of the attack methods remains invariant. CAVALRY consistently achieves the highest SRR across all three diverse evaluators, outperforming the strongest baselines regardless of the specific judge employed. This consistency confirms that our attack effectively disrupts the target model's semantic processing capabilities in an objective, evaluator-agnostic manner.

---

> ### Author Response · Authors · 2025-11-23
>
> **[Part 2/2]**
>
> ### 2. Ablation Study and Hyperparameter Sensitivity
>
> > **Comment:** *"Equation (8) introduces \lambda_1, \lambda_2, \lambda_3, but no individual ablation for each loss. Lack of sensitivity analysis limits understanding of the contribution balance... Could authors provide some insights on how to tune these parameters?"*
>
> **Response:**
> We thank the reviewer for pointing out this missing piece of analysis. We fully agree that understanding the individual contribution of each loss component and the sensitivity of their hyperparameters is crucial for reproducing and validating our framework.
>
> To address this, we have included a comprehensive parameter sensitivity analysis in the newly added **Appendix C**. As illustrated in **Figure 8**, we systematically varied each hyperparameter on the QwenVL-2.5-7B model using the stage-2 generator. The results reveal distinct roles for each term:
>
> * **Semantic Weight ($\lambda_1$):** This term is foundational. Our analysis shows a distinct performance peak at 0.1, whereas setting $\lambda_1=0$ leads to a collapse in attack effectiveness, confirming that the generative likelihood guidance is indispensable.
> * **Visual Feature Weight ($\lambda_2$):** Performance improves sharply as the value increases from 0 to 20 and stabilizes thereafter, suggesting that a higher weight is necessary to create sufficient disturbance in the visual embedding space.
> * **Auxiliary Weight ($\lambda_3$):** This term exhibits a "sweet spot" at 10, confirming that moderate regularization from the auxiliary network is essential for maintaining the transferability and spatiotemporal consistency required for video MLLMs.
>
> The significant difference in magnitude between these optimal values (e.g., 0.1 vs. 20) effectively balances the gradient scales, ensuring that the cross-entropy-based semantic loss is not overwhelmed by the L2-norm-based feature losses during optimization.
>
> ### 3. Defense Mechanisms and Model Robustness
>
> > **Comment:** *"No adversarial detection, adversarial training, or model-side robustness evaluation provided. It would be good to include some experiments on mitigation methods."*
>
> **Response:**
> We appreciate the reviewer's suggestion to rigorously evaluate potential mitigation strategies. Regarding model-side defenses such as adversarial training, we respectfully note that retraining multi-billion parameter MLLMs is computationally prohibitive for academic research and technically impossible for proprietary black-box APIs like GPT-4 and Gemini. Consequently, input sanitization (data-side defense) remains the most viable and widely deployed defense strategy for such systems.
>
> It is worth noting that our original experimental setup was designed with this reality in mind; we applied standard JPEG compression (Quality=75) by default to all adversarial videos to simulate real-world web transmission, establishing a baseline level of robustness.
>
> To fully address your request and stress-test our method further, we conducted extensive experiments in the newly added **Section 6**, employing aggressive mitigation techniques including heavy JPEG compression ($Q=50$), Gaussian Blur ($\sigma=1.0$), Median Filtering, and Bit-Depth Reduction against the state-of-the-art **Gemini 2.5**. The results are compelling: even under these techniques, CAVALRY maintains an Attack Success Rate of approximately **50%** in content moderation evasion. This persistence confirms that our generator learns robust, spatiotemporally coherent features rather than fragile high-frequency noise that can be easily filtered out by standard defenses.
>
> ---
>
> We hope these additional experiments and analyses fully resolve your concerns.

---

### Meta-Review · Area_Chair_th9e · 2026-01-07

**Summary:**

This work presents a universal adversarial attack framework applicable to both image and video MLLMs, and demonstrates the effectiveness of the method on multiple models. The reviewers' concerns primarily focus on the selection of evaluation metrics, the novelty of the proposed method, and the superiority and reliability of the experimental results. After reviewing the reviewers' comments and the authors' responses, the AC believes that, despite the authors' explanations in their response regarding how their method differs from existing approaches, the paper still lacks sufficient novelty. Additionally, the baseline methods in the authors' experiments have not been fully ablated under the experimental conditions, such as experiments where only the loss function is replaced. While the authors enrich the paper by incorporating video MLLMs and temporal techniques, this addition, although it enhances the comprehensiveness of the work, does not align well with the proposed unified framework. Furthermore, the authors did not provide a direct quantitative explanation for choosing ReaNet as the classifier instead of a transformer. Taking all factors into account, the AC concludes that although the work extends existing research, it does not provide sufficient novelty and contribution. Therefore, the AC is inclined to reject this paper.

**Reviewer Concerns:**

- 1Yvi:
The reviewer raised concerns about the domain of conflict, which has been excluded from consideration.


- wEJt:
Regarding the use of temporal mechanisms, the authors provided a reasonable explanation and addressed the reviewer's concerns.
The authors also supplied additional qualitative experimental results that resolved the question about visual distortion.
Furthermore, the authors presented ablation studies on the loss function components, evaluation results on other benchmarks, and ablation experiments using different perturbation rates, effectively addressing the relevant issues.

- Tbio:
Regarding the novelty of the work, the authors provided explanations, but still could not fully demonstrate the paper's sufficient novelty.
In response to concerns about the evaluation metrics, the authors offered corresponding clarifications and additional experimental results, which completely addressed the reviewer's questions.
On the mechanism behind transferability, the authors provided explanations and citations that fully answered the reviewer's concerns.
Regarding the choice of additional model architectures, while the authors provided references and qualitative explanations, they lacked reliable ablation studies and quantitative results, which left the reviewer's concerns unresolved.
The issue of choosing SRR as the evaluation metric was addressed.



- AP9G:
The authors provided a reasonable explanation for the use of temporal mechanisms and resolved the reviewer's concerns.
Regarding the novelty of the loss function, although the authors provided an explanation, they could not fully demonstrate that the work offered significant improvements over existing approaches, thus leaving the reviewer's doubts unresolved.
Regarding potential statistical errors in the experimental results, the authors provided additional experimental results, but these only covered part of the experimental settings mentioned in the main text, and therefore could not be considered a complete resolution of the reviewer's concerns.

- pStB:
The authors offered further clarifications to address the questions about using a generative paradigm.
Regarding the differences in loss functions, although the authors provided qualitative explanations, they were unable to sufficiently prove that the work offered more substantial and critical improvements.
The authors explained the selection of baselines and benchmarks, resolving the associated concerns.
Lastly, the authors provided additional experimental results that addressed the concerns regarding efficiency.

**Reviewer Scores:**

- 6 (domain of conflict)

- 4->6

- 2->2

- 4->4

- 4->6

---

### Decision · Program_Chairs · 2026-01-26

Reject